# Integrating molecular, histopathological, neuroimaging and clinical neuroscience data with *NeuroPM-box*

Yasser Iturria-Medina [1,2,3 ✉], Félix Carbonell[4], Atousa Assadi[1,2,3], Quadri Adewale[1,2,3], Ahmed F. Khan[1,2,3], Tobias R. Baumeister[1,2,3] & Lazaro Sanchez-Rodriguez[1,2,3]

Understanding and treating heterogeneous brain disorders requires specialized techniques spanning genetics, proteomics, and neuroimaging. Designed to meet this need, *NeuroPM-box* is a user-friendly, open-access, multi-tool cross-platform software capable of characterizing multiscale and multifactorial neuropathological mechanisms. Using advanced analytical modeling for molecular, histopathological, brain-imaging and/or clinical evaluations, this framework has multiple applications, validated here with synthetic ($N > 2900$), in-vivo ($N = 911$) and post-mortem ($N = 736$) neurodegenerative data, and including the ability to characterize: (i) the series of sequential states (genetic, histopathological, imaging or clinical alterations) covering decades of disease progression, (ii) concurrent intra-brain spreading of pathological factors (e.g., amyloid, tau and alpha-synuclein proteins), (iii) synergistic interactions between multiple biological factors (e.g., toxic tau effects on brain atrophy), and (iv) biologically-defined patient stratification based on disease heterogeneity and/or therapeutic needs. This freely available toolbox (neuropm-lab.com/neuropm-box.html) could contribute significantly to a better understanding of complex brain processes and accelerating the implementation of Precision Medicine in Neurology.

[1] Neurology and Neurosurgery Department, Montreal Neurological Institute, Montreal, Canada. [2] McConnell Brain Imaging Centre, Montreal Neurological Institute, Montreal, Canada. [3] Ludmer Centre for Neuroinformatics & Mental Health, Montreal, Canada. [4] Biospective Inc., Montreal, Canada. ✉email: yasser.iturriamedina@mcgill.ca

Most prevalent neurological disorders are highly complex, involving a continuum of biological alterations from the molecular to macroscopic (system) level. For example, Alzheimer's disease (AD), the most common form of dementia, is characterized by concurrent disruptions in genes, molecular pathways, proteins, vascularity, synapses, neuronal populations, and high-order neuronal networks[1]. The continuous crosstalk between these and other factors, as opposed to a single dominant factor, are what causes AD's associated alterations in memory, thinking, and behavior[2]. The massive failing of single-target therapeutic interventions for AD clearly demonstrates that we cannot understand nor eventually cure complex multilevel brain disorders without a deeper study of their numerous interrelating components. In keeping with the tenets of Personalized Medicine (PM), and contrary to the one-treatment-fits-all approach, treatments also need to be tailored to multiscale and multifactorial brain mechanisms as well as each individual's capacity to response[3–6].

Over the last decades, the scientific community has moved closer to understanding the imperative for an integrative (multilevel) analysis of both the brain's reorganization and associated disorders. Systems biology aims to generate spatiotemporal mechanistic models of hierarchical biological networks and the adaptive changes in the brain as it moves from a normal to a pathological condition[2,7].

The neuroinformatic field is similarly devoted to the development of analytical and computational models for the sharing, integration, and analysis of multimodal neuroscience data[8–11]. However, despite their potential to provide a better understanding of complex neuropathological processes and the individually-tailored selection of treatments, most associated methods (e.g. for separated or integrated molecular–neuroimaging analysis, data-driven patients stratification, and intra-brain spreading of pathological alterations) remain difficult to apply even when computational codes are shared, usually requiring advanced programming/technical expertise and, in many cases, even the collaboration of the developers. Simply put, vital user-friendly open access tools for both multiscale and multifactorial brain research are still lacking. Their absence is accentuated by the accelerated development of innovative approaches requiring these types of tools. This contributes to statistical inconsistencies, consumes valuable research funding, and remains a major impediment to reproducibility in research.

Motivated by these concerns, we embarked on a long-term initiative to develop, validate, and share integrative analytical modeling of molecular, histopathological, neuroimaging, cognitive/behavioral, and/or therapeutic data to advance understanding of brain (dis)organization mechanisms, at the individual and group level, as well as to identify personalized therapeutic needs. We have subsequently developed a user-friendly software substantially improving and unifying multiple methods[12–16] in a single application: the *Neuroinformatics for Personalized Medicine* toolbox (*NeuroPM-box*; Fig. 1 and Table 1). *NeuroPM-box* can be applied to any type of neuroscience data without restrictions. For example, each of these tools have been extensively tested and validated in the neurodegenerative context, but they are equally applicable to characterizing multifactorial processes in healthy neurodevelopment and aging, or in psychological disorders. Most of the outputs from the tools are biologically interpretable and the 4D-viewer enables the visualization of the brain's multifactorial spatiotemporal dynamics (e.g., tau and amyloid-β spreading through the cortex). Moreover, *NeuroPM-box* is not a static application; it was designed to be continuously expanded with new, more integrative methods to accelerate understanding of abnormal brain mechanisms and advancing the implementation of personalized care in neurology.

## Results

*NeuroPM-box* (Fig. 1) enables both the separate and combined analysis of large-scale molecular and macroscopic data, including molecular screening (transcriptomics, proteomics, epigenomics), histopathology (post mortem neuropathology), molecular imaging (amyloid, tau-positron emission tomography (PET)), magnetic resonance imaging (MRI), and cognitive/clinical evaluations. In particularly, it focuses on clarifying crucial mechanistic questions on how the brain functions; such as (i) Which series of sequential molecular or macroscopic states (e.g., genetic and brain regional alterations, respectively) underlie decades of neuropathological evolution[16,17]? (ii) Which genes (or molecular pathways) drive dysfunction in other genes and pathways[9,16,18]? (iii) How do disease agents (e.g., toxic tau and amyloid-β proteins) spread through communicating cells in the brain[12,13,19]? (iv) Which multifactorial, synergistic (causal) interactions occur in diseased brain regions[14,20]? (iv) How would each patient potentially respond to different therapeutic interventions[15]?

Each of the included tools and analytical methods were developed and successfully validated in multiple studies[12–16]. The *NeuroPM-box* aggregates considerably improved versions of these tools, and, notably, allows for the first time the synergistic combination of the different methods and data modalities for a unifying multiscale and multifactorial brain analysis. Figure 1e and Table 1 summarizes the different algorithms and provides examples of their potential for combined analysis. In brief, the user has access to four main analytical frameworks, combinations thereof, and a versatile visualization tool, described in the following subsections.

### Trajectories in large-scale molecular, imaging, and/or clinical data. 
The *contrasted Trajectories Inference* (cTI) algorithm[16] (Fig. 2; cTI definition in "Online Methods") uses recent advances in artificial intelligence (AI) to explore and visualize high-dimensional data[21] to elucidate the distinctive/contrasted underlying paths, using large-scale biological observations (e.g., genetics and neuroimaging data covering a biological process of interest, such as neurodevelopment or neurodegeneration). For example, when applied[16] to post mortem microarray gene expression (GE) data from the blood of 744 subjects on the Alzheimer's disease spectrum (ADNI data), the cTI algorithm automatically identified the series of sequential molecular states (e.g., genetic alterations) covering decades of disease progression and, subsequently, detecting the relative ordering of individuals aligned with these patterns. A molecular-disease score per subject is obtained, reflecting the individual position on the identified long-term disease "timeline". Similarly, when evaluated in 1225 post mortem brains on the AD and Huntington's disease (HD) spectrums (HD; ROSMAP and Harvard Brain Tissue Resource Center [HBTRC]), cTI strongly predicted neuropathological severity and comorbidity (Braak, Amyloid and Vonsattel stages; see Fig. 2d–e for results in HBTRC).

In the HBTRC data, we observed (Fig. 2e) a positive association between the individual molecular-disease score and the levels of neuropathologic affectation in AD and HD, which was consistent with findings in ADNI and ROSMAP[16]. The GE scores significantly associated with both the Braak stages (Fig. 2e (top); $F = 11.17$, $P < 0.001$, FWE-corrected) and the Vonsattel stages (Fig. 2e (bottom); $F = 9.04$, $P < 0.001$, FWE-corrected).

For characterizing disease heterogeneity, the cTI can also assign subjects to different subtrajectories in the contrasted space. These subtrajectories reflect different tendencies in the contrasted data, such as different disease variants. To validate the cTI algorithm's capacity to distinguish between neurological

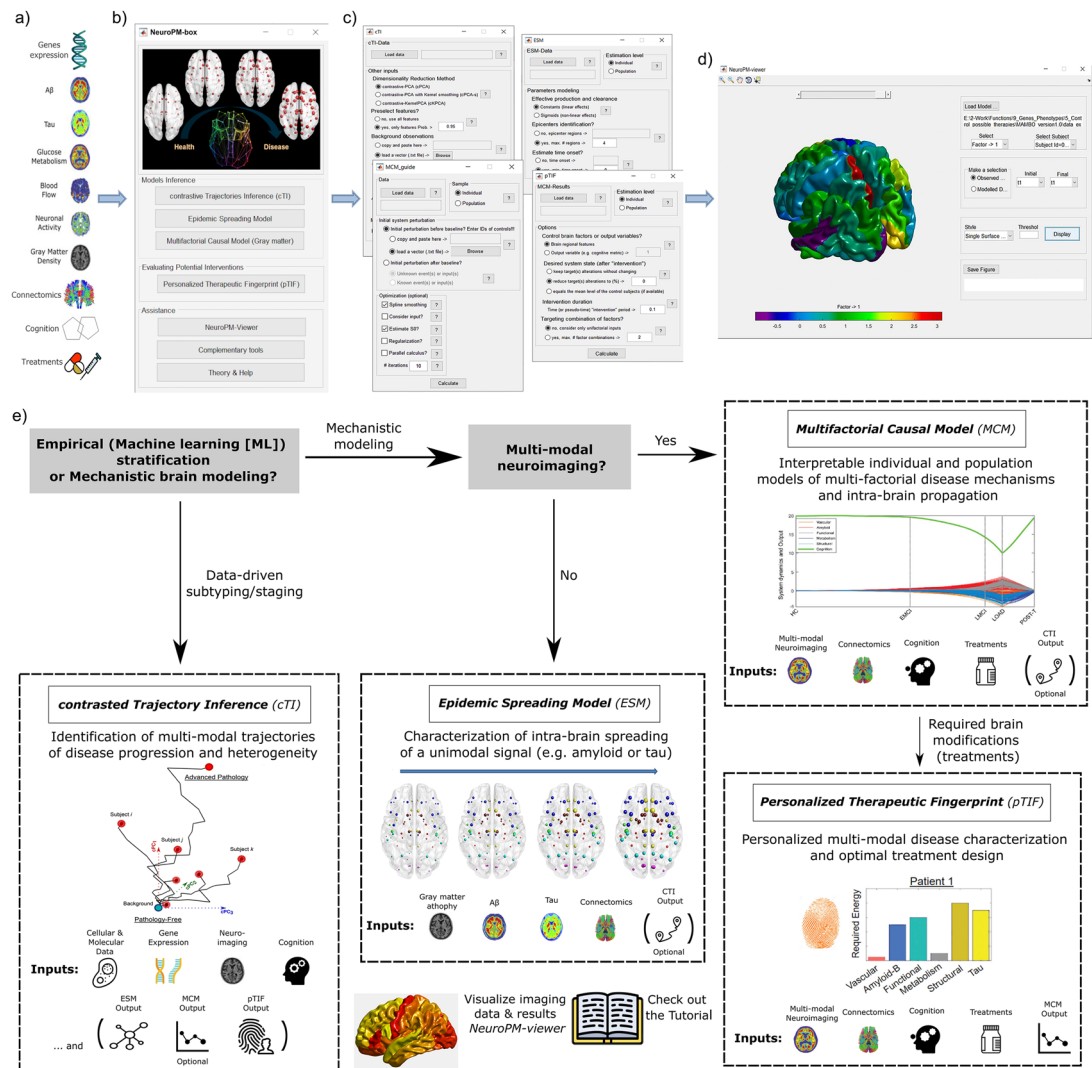

**Fig. 1 Schematic for *NeuroPM*-box software workflow and practical guidelines. a** The primary software data inputs include molecular (e.g., RNA and proteins concentration arrays), multimodal imaging (e.g., tau, amyloid-β and glucose metabolism PET, vascular, functional, and structural MRI), whole-brain connectomics (e.g., structural and vascular networks), cognitive/clinical evaluations, and therapeutic interventions (e.g. medication). There is no restriction on the number of modalities that can be analyzed. **b** The *NeuroPM*-box interface allows users to select from four analytical methods (tools), apply auxiliary applications, and access the visualization tool (*NeuroPM-viewer*) and software tutorial. **c** Main software modules supporting the data-driven analysis of the multimodal data. **d** *NeuroPM-viewer* enables detailed exploration of the human cortex of both real and modeled spatiotemporal brain dynamics (see also Fig. S2). **e** Practical guidelines for methods users (available methods are further described in Table 1 and subsections below). Essentially, the analytical methods belong to two main categories, empirical and mechanistic. The former is purely data driven and focus on identifying and interpreting intrinsic patterns in the data without making strong a priori biological assumptions. Specifically, the included algorithm (see *contrasted Trajectory Inference* subsection and summary on Table 1) provide individualized quantitative scores reflective of disease progression and assign each subject to distinctive subpopulations (tentatively reflecting different disease subtrajectories). Any type of quantitative data can be used as input (e.g. transcriptomic, proteomic, histopathological, metabolomics, multimodal imaging, clinical), while each data-feature's contribution to the subjects' final stratification is quantified, revealing the most informative features (e.g. specific genes, brain regions, clinical evaluations) and associated data modalities (e.g. RNA, imaging, clinical). However, the user should avoid performing causal interpretations based on empirical modeling, because the intrinsic limitation to distinguish between direct and indirect biological effects. Mechanistic models, by the contrary, aims to decode cause-effects in terms of biological factors alterations spreading through physical brain connections and/or synergistic factor–factor interactions contributing to spatiotemporal brain reorganization. The two implemented generative models focus on uni-modal or multimodal imaging data, i.e. ESM considers the intra-brain spreading of a unique biological factor measured with an specific imaging modality (e.g. tau-PET, or amyloid-β PET), while MCM considers the direct (causal) interactions and concurrent intra-brain spreading of multiple biological factors' alterations quantified with different imaging modalities (tau, amyloid-β, and glucose metabolism PET; cerebrovascular flow, functional activity indicators, and structural atrophy measured with MRI). Notably, mechanistic approaches (ESM, MCM, pTIF) can be informed by the empirical data-driven outputs (cTI stratification), allowing the incorporation of a wide range of possible multiscale biological information (e.g. molecular and clinical stages and subtypes) on the imaging-based generative brain models (see "cTI → ESM, MCM, pTIF" method on Table 1). Finally, personalized causal brain models identified by the multimodal mechanistic approach (see MCM) can be interrogated to identify individual therapeutic needs in terms of biological deformations required to stop/revert factor(s)-specific (imaging modalities) alterations or clinical deterioration (see pTIF on Table 1 and subsequent subsection).

**Table 1 Main *NeuroPM-box* approaches and their synergy.**

| Class | Method | Required data | Description | Applications and novelty |
|---|---|---|---|---|
| A, MInt | cTI | Transcriptomics, proteomics, epigenomics, histopathological, brain imaging, and/or clinical | Identifies, in large-scale biological observations, the distinctive underlying pathways/patterns covering a biological process of interest (e.g., large-scale genetics and neuroimaging data collected during neurodevelopmental or neurodegenerative evolution) | Biologically defined patients stratification: identification of subpopulations presenting potentially different disease variants, and staging of individuals within each subpopulation according to the pathological severity. Detection of the most informative features and data modalities (e.g., genes, proteins, imaging, or clinical biomarkers) for tracking the studied disease process. Narrowing the focus to most-informative-data modalities will considerably reduce the time and cost of patient evaluations in clinic settings |
| A | ESM | Brain imaging | Characterizes the intra-brain propagation of an infection-like "agent" through physical brain networks, estimating agent production and clearance rates, initial topological distribution in the brain, and onset time | Analyzing the spatiotemporal intra-brain spreading of disease factors through brain networks (e.g. misfolded proteins propagation) |
| A, MInt | MCM | Brain imaging, clinical, treatment | Characterizes direct multifactorial brain interactions (e.g., how toxic misfolded proteins alter cerebral blood flow, and vice versa), concurrent "agents" spreading through physical networks (e.g., misfolded proteins and neuronal alterations propagation through axonal and vascular connections), subsequent impact on cognitive/clinical integrity, and external input effects (e.g., impact of drug or physical exercise) | Mechanistic characterization of healthy and diseased brain (dis)organization processes by integrating multimodal neuroimaging, cognitive, and therapeutic data at the individual or population level Testing and comparing multiple disease hypotheses, by detecting the driving/dominating biological factors, initial regional epicenters, and/or most-likely disease triggering events (e.g., an abnormal misfolded proteins accumulation, a vascular or neuronal perturbation) |
| A, MInt | pTIF | Brain imaging, clinical | Estimates the biological modifications required to conduce the brain from a given state (e.g., diseased) to a desired state (e.g., healthy) | Prediction of personalized therapeutic needs and treatments effectiveness by analyzing how different brain factors interact and would respond (at the individual level) to potential clinical perturbations. Stratification of patients according to treatment needs |
| A, MInt | cTI → ESM, MCM, pTIF | From subject stratification to imaging | Mimics longitudinal datasets for fitting ESM and MCM models at the population level by ordered and concatenated subjects in ESM and MCM files according to cTI outputs (pseudo-times and subtrajectories) | Integration of molecular, histopathological, and neuroimaging factors using molecular-based cTI outputs (e.g., from gene expression arrays) for the reordering of subjects into imaging-based ESM, MCM, and pTIF optimizations. Dynamic analysis of population-based cross-sectional neuroimaging data using the temporal information identified by cTI for pseudo-longitudinal ESM, MCM and pTIF optimizations |
| V | NeuroPM-viewer | Brain imaging | Visualizes real and simulated imaging data over the cortex, with customizable options for brain views/planes and temporal windows of interest | Detailed visual exploration of concurrent multifactorial brain spatiotemporal patterns |

Methods: contrasted Trajectories Inference (cTI), multifactorial causal model of brain (dis)organization (MCM), epidemic spreading model (ESM), and personalized therapeutic intervention fingerprint (pTIF). Each method includes a detailed tutorial that illustrates how the applications/modules can be combined to undertake complementary analyses. Potential applications are classified as data analysis (A), multimodal integration (MInt), and data visualization (V).

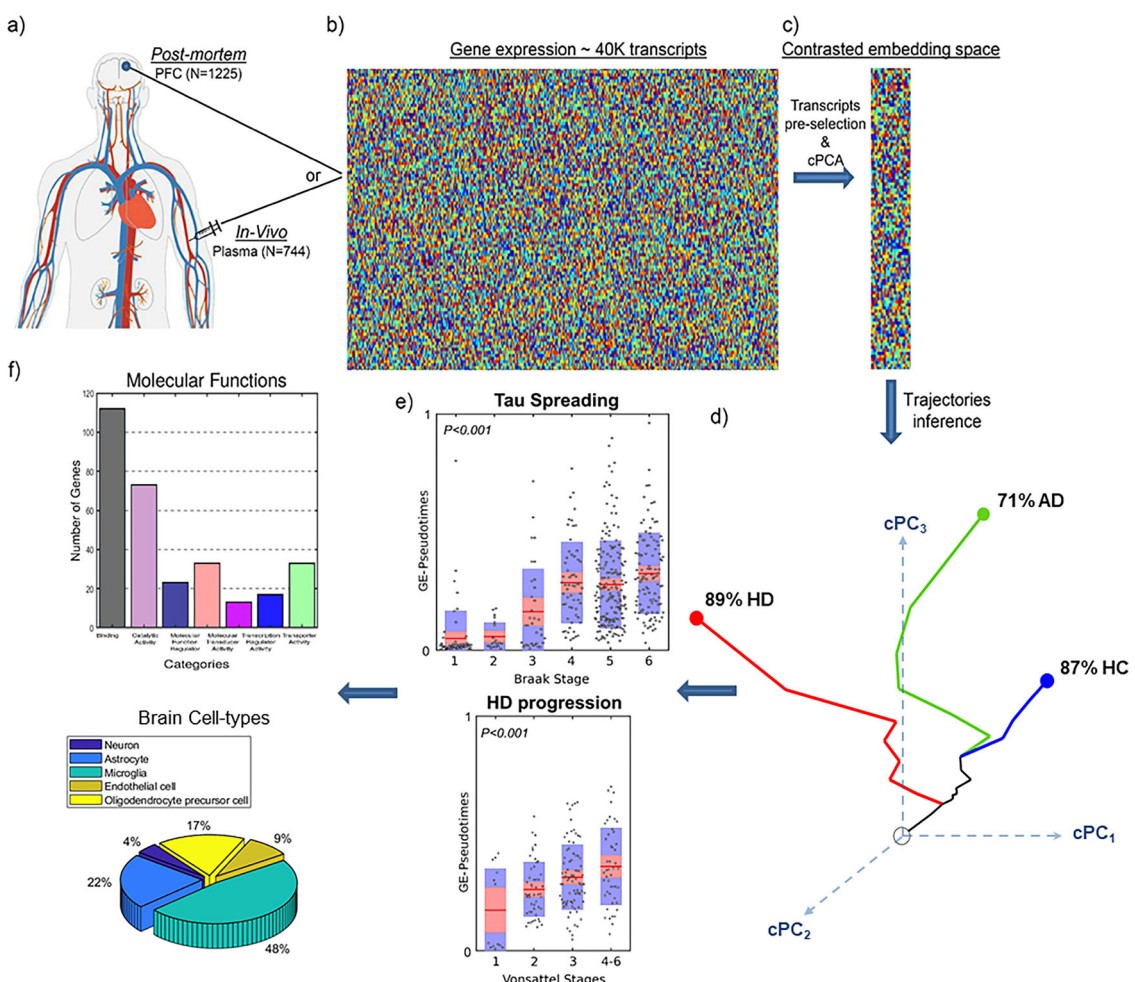

**Fig. 2 Schematic of *cTI* application to detect disease-associated patterns and patient neuropathological stages in neurodegeneration[16]. a** In vivo blood ($N = 744$; ADNI) and post mortem brain ($N = 1225$; ROSMAP, HBTRC) tissues, from Alzheimer's disease (AD), Huntington's disease (HD), and/or normal controls (HC) subjects, are screened to measure the activity of ~40,000 transcripts. **b** Each population's high-dimensional data are reduced to a set of disease-associated components via **c** contrastive principal component analysis (cPCA)[21,62]. **d** This allows each subject to be represented in a reduced *n*-dimensional disease-associated space where the corresponding position reflects his/her pathological state (proximity to the bottom-left corner implies a pathology-free state; conversely, the top-right corner implies advanced pathology). For instance, when analyzing the GE data from the HBTRC's highly heterogeneous population (including HC, LOAD, and HD subjects, total $N = 736$), the high-dimensional data were reduced to seven contrasted PCs [cPCs] capturing up to 97.5% of the population variance (and individually explaining 38.73%, 19.91%, 16.18%, 8.46%, 5.85%, 5.50%, and 2.87% of the variance, respectively). Notice that, for visualization simplicity, here were only represented the first three cPCs, but the quantitative analysis considers all identified components. Within this cPCs space, each subject is automatically assigned to a disease trajectory that represents a subpopulation of subjects potentially following a common disease variant (see "Methods"). The number of subpopulations (disease trajectories) is determined automatically based on how the subjects "cluster" together in the disease-associated space. **e** An individual molecular-disease score is then calculated, reflecting how advanced each subject is in his/her disease trajectory. This score significantly predicts neuropathological deterioration. **f** Finally, the resulting model weights (from contrastive PCA) allow the identification and posterior functional analysis of most influential genes/features. Panels **a–c**, **e**, **f** adapted with permission from ref. [16].

conditions in highly heterogeneous populations, here we separately reanalyzed the GE and histopathological data from the HBTRC, including two disorders (late-onset AD [LOAD] and HD) and nondemented controls (Dataset 1, $N = 736$; "Online methods"). Using each data modality (GE or histopathology), the cTI method automatically identified multiple subtrajectories reflecting diagnosis-specific subpopulations (Fig. 2d). For instance, based on GE, sub-trajectory 1 comprised 87% of the nondemented controls, 18% of the AD subjects, and 2.7% of the HD subjects; sub-trajectory 2 comprised 71% of the AD subjects, 38% of the controls, and 12% of the HD subjects; and sub-trajectory 3 comprised 89% of the HD subjects, 32% of controls, and 25% of the AD subjects.

Similarly, based on a limited set of histopathological data (just 25 broad atrophy metrics; see Example Dataset 1), sub-trajectory

1 comprised 91% of the controls, 28% of AD subjects, and 4.9% of HD subjects; sub-trajectory 2 comprised 62% of the HD subjects, 57% of the AD subjects, and 40% of the controls; and sub-trajectory 3 comprised 48% of the HD subjects, 26% of the AD subjects, and 2.3% of the controls.

Furthermore, we used synthetic data ($N > 2900$) to extensively test the cTI's performance under different population characteristics, specifically, with different levels of population heterogeneity and sample sizes. A simulation study (Fig. S1 and demo provided with software) confirmed the model's capacity to accurately identify individual disease stages and subtypes, as well as to recover the biomarkers' contributions to the predictions, in the presence of noisy data. Altogether, the results with synthetic and real data demonstrate that cTI is a promising technique for patient stratification in terms of disease stages and variants, even when

comorbid neurological conditions and noisy observations are considered. In addition, cTI can potentially be used to extract intrinsic dynamic information from large-scale, cross-sectional data. As described in Table 1 (Application $cTI \rightarrow ESM, MCM, pTIF$), this functionality can be particularly useful in analyzing cross-sectional neuroimaging studies as if they were longitudinal studies. That is, individual pseudo-times and subtrajectories from cTI (or any user-provided patient stratifications[10,22]) can be used to mimic longitudinal datasets for adapting ESM, MCM, and pTIF models to subpopulation levels (Tutorial and/or Text S1).

**Epidemic spreading of pathological agents.** The epidemic spreading model (ESM) in the neurological context[12] characterizes the intra-brain propagation of infection-like "agents" (e.g. misfolded proteins [tau, amyloid, alpha-synuclein, TDP-43]) through physical brain networks (e.g., anatomical, vascular, functional). The ESM (Fig. 3a, b) estimates individual rates of "agent" clearance and production, which can follow a linear or sigmoid relationship with the local concentration of the modeled infection-like "agent".

ESM has been successfully applied to further understand the spread of toxic amyloid and tau proteins in the neurodegenerative human brain[12,13]. Importantly, the *NeuroPM-box* version of the ESM model presents five main significant improvements compared to the initial model applications[12,13]:

(i) Mathematical extension to work with the direct imaging signals (e.g., SUVr values from PET) or with probabilistically inferred values from the images (the original model was defined only for probabilistic values).

(ii) By considering all available time points, the improved ESM optimization can focus on individual- and population-based longitudinal data (the original model was evaluated only for cross-sectional data), thereby increasing the robustness and biological interpretably of the model's estimated parameters.

(iii) ESM now covers most of the possible numeric values for parameter optimization, using a considerably more robust algorithm (MATLAB's *MultiStart*[23]) to solve the non-linear differential equations. Specifically, gradient-based solvers are applied to find local minima from multiple starting points in search of global minima solutions (instead of potential local minimums as with the initial implementation). This modification effectively improves robustness and interpretably of the estimated biological parameters.

(iv) Epidemic production and clearance rates can be defined either as linear or exponential/sigmoid functions (optional), increasing the flexibility of the basic formulation by considering both linear and non-linear biological processes at the regional level (the initial model was considering only exponential production and clearance rates).

(v) Considering the influence of different physiological factors and/or random noise on the analyzed imaging modality and the fact that a non-zero regional value does not necessarily imply the presence of the studied "agent" (e.g. amyloid or tau deposition) but just background fluctuations on the image signal. Subsequently, for estimating the regional epicenters, the improved ESM algorithm allows the definition of a maximum (non-zero) value below which the regions are still considered free of "agent" presence but with their typical background 'noise' (e.g. tau and amyloid positivity thresholds). Only regions over this value are considered as likely epicenters. A notable improvement in model fit has been observed for both tau- and amyloid-PET spreading analysis.

In Fig. 3, we show the application of the improved ESM algorithm to a healthy and diseased population of 105 participants, each having at least two longitudinal tau-PET acquisitions ($^{18}$F-AV1451 ligand, ADNI data; "Online methods"). On average, when applied at the individual level, this approach explained 80% (SD = 9.6, all $P < 10^{-6}$) of the variance in regional tau values across all available time points and subjects. Figure 3c–e shows the results of the EMS for a clinically healthy female control participant with significant memory complaints and four longitudinal tau-PET evaluations (subject ID 024_S_5290 in ADNI). Starting from a pathology-free stage, the model explained 86% ($P < 10^{-10}$) of the variance in tau values across the four available time points (Fig. 3c, d). Notice the strong correspondence between the observed and reproduced tau deposition patterns at the first and last PET evaluations, ages 71 (Fig. 3c, right) and 73 (Fig. 3d, right) respectively. In addition, the ESM automatically identified the entorhinal cortex, fusiform gyrus, caudate anterior cingulate, and inferior parietal in the left hemisphere as the most-likely epicenters for tau spreading in this subject, and estimated that tau accumulation and spreading started around age 62. Figure 3e illustrates the long-term intra-brain propagation process, starting with the identified epicenters and diffusing, over a decade, to the other brain regions, following a stereotypic AD-related pattern[24]. In this analysis, we allowed up to a maximum of four regions as initial epicenters and a 5% of the maximum observed SUVr for the starting values of the non-epicenter regions at the onset time.

**Multifactorial causal model of brain (dis)organization.** The multifactorial causal model of brain (dis)organization and cognition (MCM[14]; Fig. 4a, b) accounts for: (i) first pathological perturbation in the disease process (the brain regions and biological factors that are initially altered), (ii) regional multifactorial causal interactions (e.g., how toxic misfolded proteins alter cerebral blood flow [CBF], how subsequent alterations in CBF influence neuronal activity and gray matter atrophy, and vice versa), (iii) concurrent propagation of perturbations through physical networks (e.g., intra-brain propagation of misfolded proteins, vascular or neuronal alterations across axonal and vascular connectomes), (iv) the subsequent impact of (i) and (ii) on cognitive/clinical integrity, and (v) the global factor-specific effects of external inputs (e.g., how a given clinical treatment impacts tau, amyloid and CBF). The MCM considers that once a factor-specific event occurs in a given brain region, or set of regions, that it can directly interact with other biological factors to alter their states. The alterations can also spread to other brain areas through physical connections (e.g., anatomical, vascular connections), where similar factor-to-factor and propagation mechanisms may occur in a continuous cycle. The MCM has been successfully applied to the study of AD[14,15], clarifying multifactorial disease-specific mechanisms, and is currently being applied to other neurodegenerative disorders (e.g., amyotrophic lateral sclerosis, frontotemporal dementia, Parkinson's disease).

Importantly, the MCM version included in *NeuroPM-box* incorporates multiple enhancements over the initial version[14], effectively improving robustness and interpretably of the estimated biological parameters. The two most remarkable enhancements are (see MCM definition in "Online methods"):

(i) The ability to focus on individual longitudinal data (in addition to group-level fitting, per the initial article) by accommodating all available time points for each individual (Eq. 12).

(ii) The differential equations are now solved in a considerably more robust way to cover most possible numeric values for parameter optimization (MATLAB's *MultiStart* algorithm[23]). Gradient-based solvers are applied to find local minima from

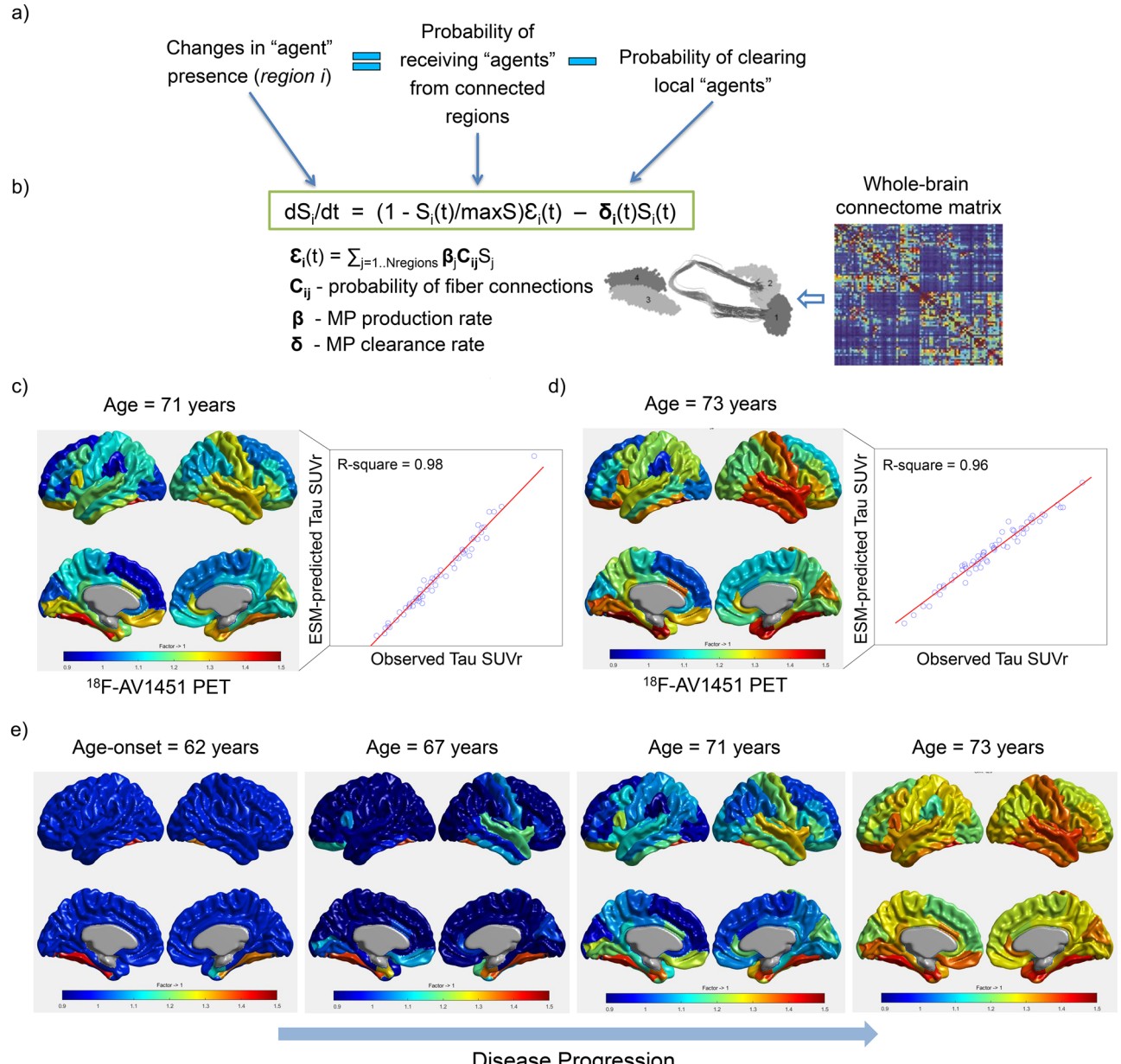

**Fig. 3 ESM approach and prediction of intra-brain tau spreading. a** Changes in the presence of a given infection-like "agent" factor (e.g., amyloid, tau misfolded proteins [MP]) at a specific brain region, "*i*", are modeled as a function of the incoming "agent" from each connected region, "*j*" (i.e. spread effects through communicating cells), minus the local "agent" clearance. **b** This dynamic cause–effect model can be mathematically translated to a non-linear system of differential equations, which is dependent on the individual "agent" production rate, the clearance rate, and the inter-region brain-connectivity matrix. In the *NeuroPM-box*, both the production and clearance rates can be optionally modeled as time-dependent sigmoid functions[12] or as global constant values. The connectivity matrix can be estimated via diffusion MRI tractography[63,64] or an alternative technique[65–67]. **c** Shows ESM results reproducing the tau deposition patterns at the first $^{18}$F-AV1451 PET evaluation (age = 71 years) of a clinically healthy female control with significant memory complaints (ADNI data, subject ID 024_S_5290). **d** ESM results in the same participant at the fourth time point evaluation (age = 73 years). Starting from a pathology-free stage, the model explained 86% ($P < 10^{-10}$) of the variance in tau values across the four available time points. **e** ESM simulation of the whole-brain intra-brain tau spreading process from the estimated onset time of tau appearance/propagation (age = 62 years) to the last observed time point. In **c**–**e**, tau values are cortical-to-cerebellum standardized uptake value ratio (SUVr).

multiple starting points in search of global minima solutions (instead of potential local minimums, as with the initial implementation).

Here, the improved MCM algorithm was applied to a healthy and diseased population of 504 participants (ADNI data, "Online methods"), each having 4–6 different imaging modalities and at least four longitudinal evaluations. The imaging modalities included tau-PET, amyloid-PET, FDG-PET (for quantifying glucose metabolism), resting-fMRI (for neuronal activity at rest), ALS-MRI (for cerebral blood flow), and structural-MRI (for gray matter density). For all subjects, the average longitudinal time window was 5.1 years (SD = 3.6). The MCM numerical optimization successfully converged in 98.4% of the population (496 subjects out of 504). Starting from a pathology-free state at the individual level, on average the MCM explained 92% (SD =

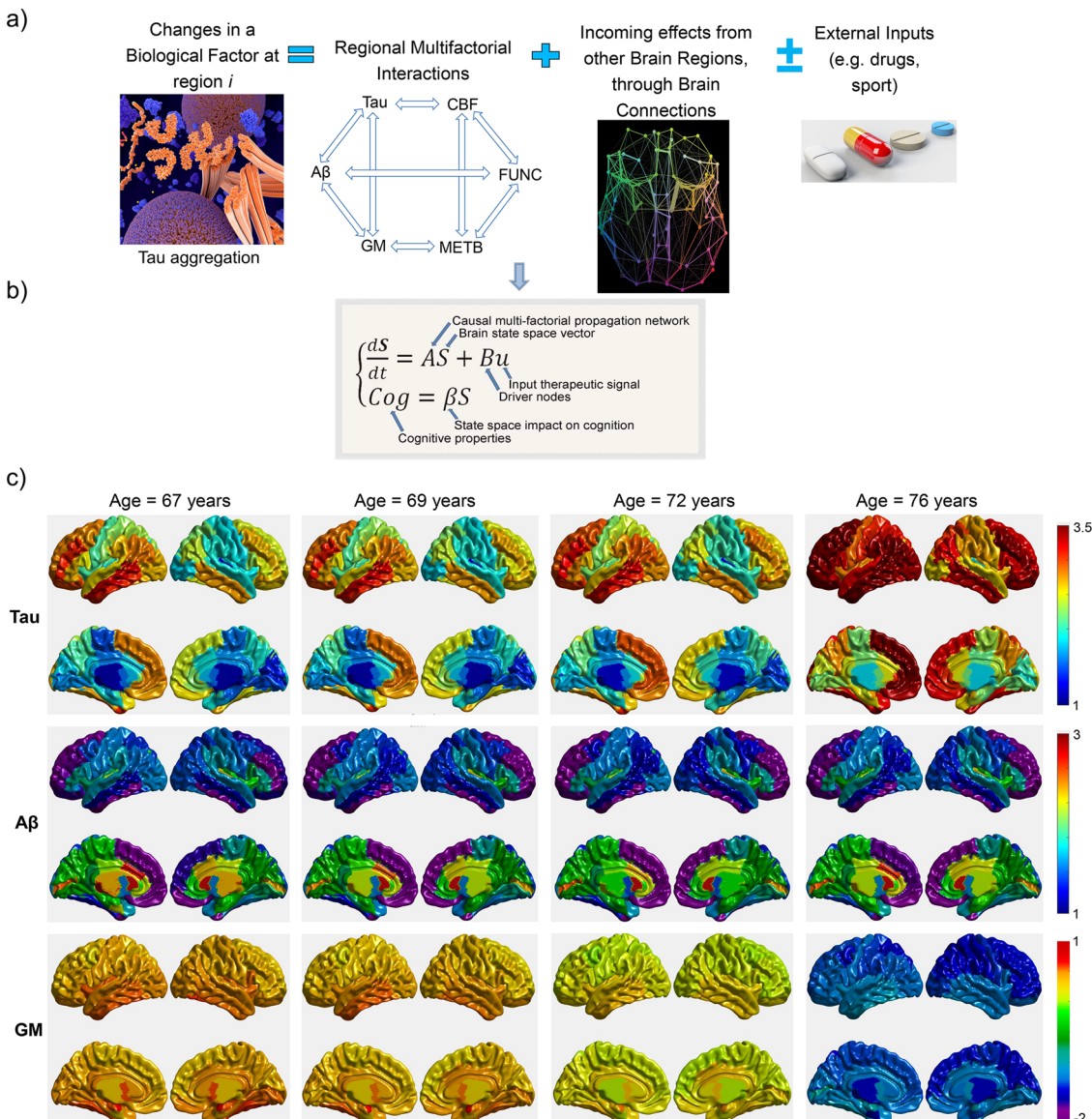

**Fig. 4 MCM definition and prediction of concurrent tau, amyloid, and structural brain changes in preclinical AD. a** Changes in a given biological factor (e.g., amyloid, tau deposition) at a specific brain region are modeled as a function of the local multifactorial synergistic interactions (e.g., how dysregulation of the cerebrovascular flow influences amyloid and tau depositions), the intra-brain alterations spreading through communicating cells, and external inputs (e.g., treatments). **b** This dynamic cause–effect model can be mathematically translated into a system of differential equations. Similar to previously proposed causal models of brain functioning[12,68–70], in MCM, causality is intrinsic to its differential equations. Beyond traditional single-factor modeling approaches (commonly neuronal activity or misfolded proteins), MCM equations also describe: (i) how the present state of a given biological factor, in a given brain region, causes new fluctuations to itself or to other biological factors in the same or a different brain region, via multifactorial local interactions or by spreading through brain connections, and (ii) how the brain's dynamic physical system could change due to the influence of external inputs (e.g., cognitive/sensory stimulus, therapeutic interventions, environmental influences). **c** MCM-simulated concurrent intra-brain changes in tau, amyloid, and gray matter (GM) density in a clinically healthy female participant with significant memory complaints (ADNI data, subject ID 024_S_5290). The resulting MCM simulation is based on the participant's actual parameters (collected from 67 to 72 years). Then, for prediction purposes, we calculated an additional time window (from 73 to 76 years) of multifactorial data. Note the prominent increase in tau brain deposition in parallel to a substantial reduction in gray matter density in the brain. In **a**, FUNC refers to functional activity at rest (e.g., from fMRI); METB refers to glucose metabolism (e.g., from FDG-PET).

4.7, all $P < 10^{-6}$) of the first and 71% (SD = 15.1, all $P < 10^{-6}$) of the last multimodal observations for all individuals, respectively. Figure 4c illustrates the novel MCM-based analysis of concurrent intra-brain changes in tau, amyloid, and gray matter atrophy on a clinically healthy female control with significant memory complaints (ADNI data, subject ID 024_S_5290). This is the same subject who was previously analyzed with ESM [Fig. 3c–e] but here is characterized from a multifactorial perspective (i.e.,

going for the first time from a univariate tau analysis to an integrative multimodal characterization). In this subject, for the first (at 67 years) and last observed time points (at 73 years), the model explained 94.6 and 51.9% of the variance, respectively, across the six data modalities. Illustrating the capacity of the MCM algorithm to predict future disease progression, data simulation (Fig. 4c) is extended for three additional years after the last time point (from 73 to 76 years). In this specific case (Fig. 4c),

we see a prominent increase in tau brain deposition in parallel to a substantial reduction in gray matter density in the brain.

**Imaging-based therapeutic fingerprints**. The *personalized Therapeutic Intervention Fingerprint* (pTIF[15]; Fig. 5) assumes that patients in a heterogeneous population require different treatments, depending on both the unifactorial alterations in their brain (e.g., tau/amyloid deposition or not, cerebrovascular alterations or not, atrophy or not) and their multifactorial brain dynamics: How different biological factors interact and would potential respond (at the individual level) to clinical perturbations. Based on spatiotemporal analysis of multimodal imaging data (PET, MRI, SPECT), pTIF values are a set of multivariate metrics that reflect the biological changes required to stop a specific brain-reorganization process or to revert the condition to a normal state. In other words, the pTIF can integrate large amounts of data (e.g., thousands of multimodal brain imaging measurements) into a simplified, patient profile—the *fingerprint* —representing the quantitative modifications of the biological factors that are needed to control the reorganization process (e.g., disease evolution) in that individual. Results using aging and late-onset AD data (ADNI) demonstrate how the pTIF algorithm can categorize patients into distinct therapy-based subtypes that correspond strongly with differential RNA profiles[15]. The multimodal imaging-derived pTIF vastly outperforms cognitive and clinical evaluations when predicting individual GE alterations. Furthermore, pTIF-identified patient subgroups present distinctively altered molecular pathways in the blood, supporting the identification of dissimilar pathological subtypes and therefore therapeutic needs in the studied population (Fig. 5c).

**Visualizing observed and modeled spatiotemporal brain dynamics**. Deeper understanding of the brain processes under study requires visualization of both acquired and simulated brain data. *NeuroPM-box* includes a versatile, user-friendly interface for visualization of all the analyzed brain factors and their dynamic changes (Fig. S2). Time is one of the most important variables for both modeling and visualization; consequently, the *NeuroPM-viewer* allows to adjust time variables (among many other settings) according to specific visualization needs.

**Versions control and stability**. Following standard-practices for sharing computational neuroscience software[25], we employ advanced source code management tools for version control (GitHub, https://github.com/). In addition, modifications performed to each software version are annotated and made available to the user in a .doc file on the software's webpage, including intuitive and technical explanations. Associated biological implications are also discussed. Furthermore, it is not only important to count with advanced informatic techniques, but also that they provide stable results. The stability of the implemented methods (cTI, ESM, MCM, pTIF) has been successfully confirmed across multiple computational workstations with both different capabilities and operational systems (*Linux, macOS, Windows*).

**NeuroPM-Box tutorial**. The software's user guide provides an in-depth explanation of all features and options available. It includes step-by-step instructions for installation (for *Linux, macOS,* and *Windows* systems), data organization-standardization, models-specific inputs and optimization, outliers' correction, data visualization, and outputs description and interpretation. The tutorial is available from *NeuroPM-box* ("Theory and Help" icon), and as a PDF from https://www.neuropm-lab.com/neuropm-box.html. See also Text S1 for easy-to-follow instructions.

Additionally, synthetic data for cTI testing/evaluation (Fig. S1) and a practical demo script are provided.

## Discussion

To the best of our knowledge, *NeuroPM-box* is the single cross-platform, open access, user-friendly software for integrating large-scale molecular, macroscopic, and clinical data using advanced mathematical modeling existing at the moment. *NeuroPM-box* allows separated and combined analysis of data derived from molecular screening (transcriptomics, proteomics, epigenomics), histopathology (post mortem neuropathology), molecular imaging (amyloid, tau-PET), macroscopic MRI, and cognitive/clinical evaluations. Most available packages focus exclusively on molecular[26–28] or brain imaging[29–33] analysis, not on their combined analysis, which *NeuroPM-box* is specifically designed to address. Moreover, no other user-friendly software includes models for characterizing the intra-brain spreading of alteration effects (e.g., connectome-mediated tau and amyloid propagation as characterized by ESM and MCM) or for identifying individual therapeutic needs based on dynamical system analysis and control theory (e.g., pTIF). Although some validated computational codes for biomarkers-based patient stratification in the neurological context have been shared[10,22], the user requires programming or technical skills to apply them. *NeuroPM-box*'s user-friendly implementation combined with recent enhancements to the methods included will accelerate the comparison, and potential integration, with several recently developed methods published by other teams[10,22,26,34].

Importantly, no advanced mathematical and/or computational knowledge is required to use the *NeuroPM-box*, as each model is described in intuitive biological terms. However, it is deliberately designed to be a post-processing analytic software, not a pre-processing package, for which many excellent free software already exist (e.g., GEPAS, Bioconductor, SPM, FSL, ANTS, CIVET, FreeSurfer, MRtrix3, DSI studio, BrainSuite). Consequently, basic molecular and imaging preprocessing (imaging registration, brain parcellation, quality control) should be completed beforehand. *NeuroPM-box* users should have basic expertise in writing/reading numerical data to text files and, when using a large-scale population, be able to organize the data into the required formats (see Text S1).

*NeuroPM-box* is a long-term, ongoing initiative. All of the tools included are under continuous development, particularly in terms of improving their numerical optimization (an open-ended field in research) and the interpretation/visualization of results. New tools and methods are also under development, with the goal of further integrating multiscale and multimodal neuroscience research. Future methodological additions will focus on continue bridging molecular, brain macroscopical factors (e.g. neuroimaging-derived biomarkers) and clinical data. For instance, we are in the process of incorporating a novel GE informed MCM approach[35,36], which proposes a general formulation that integrates whole-brain transcriptomic data of hundreds of landmark genes with multiple neuroimaging-derived biological factors (i.e. amyloid, metabolic, and tau-PET; vascular, functional, and structural MRI) and individual cognitive/clinical information. This unifying method, successfully validated on healthy aging and AD populations, concurrently accounts for the direct (causal) influence of hundreds of genes on regional macroscopic multifactorial effects, the pathological spreading of the ensuing aberrations (tau, amyloid) across axonal and vascular networks, and the resultant effects of these alterations on cognitive/clinical integrity. A similar multiscale brain model integrating neurotransmitter receptor densities with multimodal neuroimaging is also under development[37]. Similar that for the

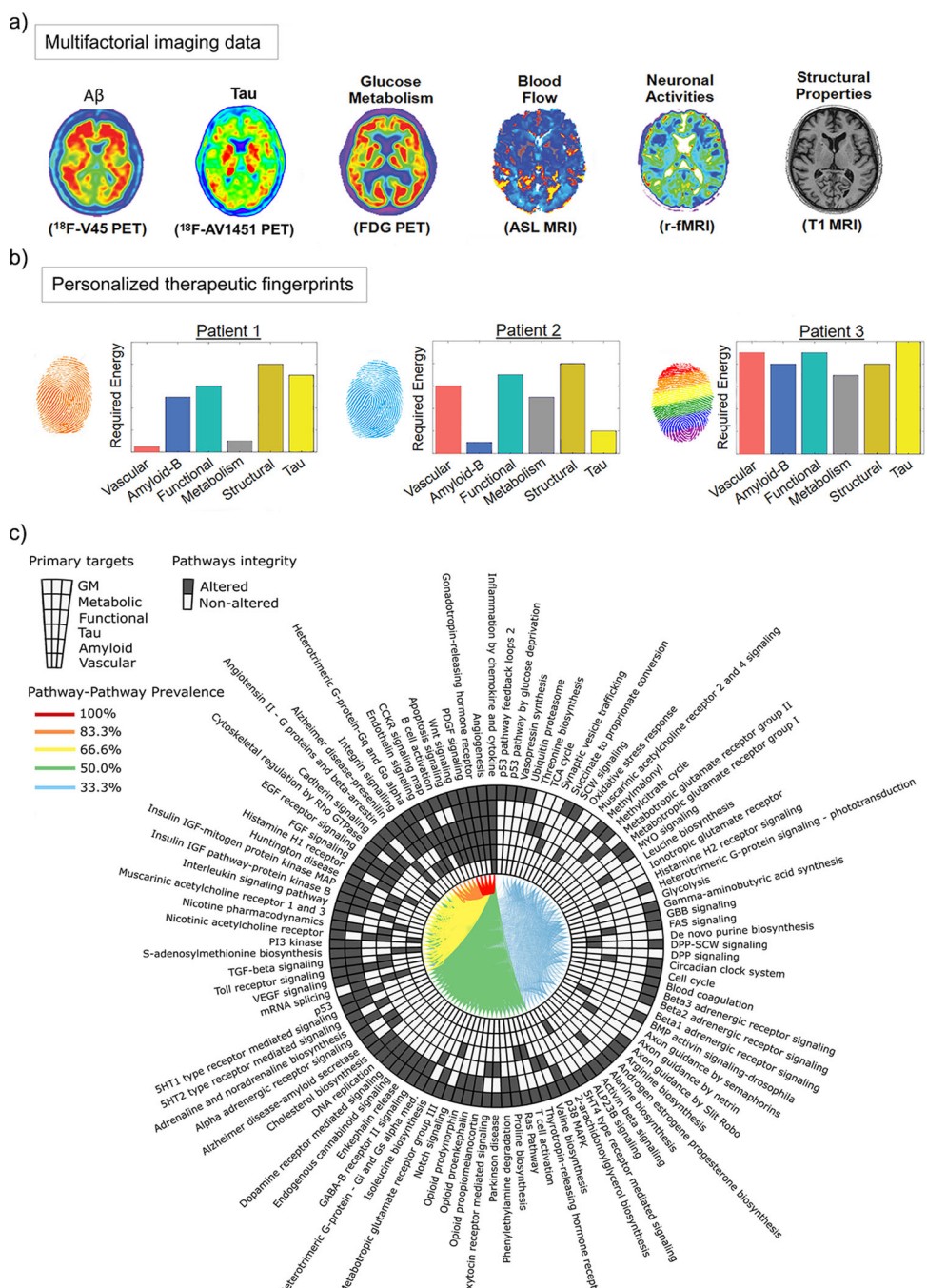

**Fig. 5 From multimodal imaging to therapeutic fingerprints and altered molecular pathways (ADNI data). a** Imaging for amyloid, tau, CBF, functional activity at rest, glucose metabolism, and gray matter density. Based on a network-based approach[14], pTIF enables individual characterization of the direct factor-to-factor intra-brain biological interactions and the multifactorial spreading mechanisms through vascular/anatomical connections. Inverting its fundamental equation provides an estimation of the changes required to produce a desired state (e.g., healthy); hence, the pTIF is defined as the set of changes required for each patient. **b** Dissimilar pTIF patterns for three participants with the same diagnosis. Patient 1 requires low-cost vascular and metabolic interventions and Patient 2 requires low-cost interventions for anti-Aß and anti-tau interventions, suggesting different single-target therapies could benefit both patients (e.g., physical exercise and aducanumab, respectively). However, Patient 3 requires multiple single-target interventions, suggesting that a high-cost combinatorial treatment, as opposed to a single-target treatment, would be more beneficial for this patient. **c** The altered molecular pathways (blood data) underlying the distinct single-target therapeutic needs. Starting at 12 o'clock and moving counter-clockwise, the pathways for each of the single-target subgroups were sorted according to prevalence. Each link between a given pair of pathways corresponds to the percentage of subgroups for which molecular pathways were found to be affected. Images (**a–c**) adapted with permission from ref. [15].

MCM approach, inclusion of spatial molecular information and estimation of external inputs effects in the ESM are planned to be incorporated in future software versions.

Our goal in sharing *NeuroPM-box* is threefold: (i) to accelerate research and clinical use, (ii) to seek feedback on any limitations and translational gaps that need to be addressed, and (iii) to further validate the *NeuroPM-box* tools, thereby increasing their applicability. It is important to emphasize that, although the cTI results presented in this article are based on specific data types (bulk transcriptomics, histopathological data), there are not restrictions in the kind and number of data modalities that can be used with this technique. Single-cell transcriptomic analysis is equally feasible with the current cTI implementation, potentially allowing the direct comparison with several trajectory inference methods originally proposed for such data type[38–41]. Furthermore, in complementary analyses, we are investigating the cTI's capacity to concurrently integrate different data modalities (molecular multi-omics, multimodal neuroimaging, and/or several cognitive/behavioral/clinical evaluations), which will be the main focus of our next studies in neurodegeneration.

To facilitate quality control when using large-scale datasets, *NeuroPM-box* currently includes specific outlier detection methods (e.g., the three sigma rule) as well as data correction and completion via imputation[42]. A common concern when applying advanced neuroscience computational techniques is runtime efficiency. Most of *NeuroPM-box*'s optimization algorithms have been implemented to minimize computational time. For instance, cTI can analyze thousands of subjects and large-scale omics data in just a few minutes. However, the differential equation-based methods (ESM, MCM) are more computationally expensive, particularly when applied at the individual level. The "default" optimization for these methods can significantly reduce the computational time, without compromising accuracy, in comparison to other available techniques (e.g. trust-region-reflective algorithm[43]). However, analyzing hundreds of subjects with multimodal longitudinal imaging data from a regular workstation could take a few days (depending on the number of modalities, brain regions, and time points; see Text S1). We are planning to upload the software to popular High-Performance Computing (HPC) portals, such as The Neuroscience Gateway (NSG, http://www.nsgportal.org) and CBRAIN (http://www.cbrain.ca). Finally, to increase the software's generalizability, we are also working to extend data input to popular organizational formats, including the *Brain Imaging Data Structure* (BIDS) standard[44].

## Online methods
### Data
*Ethics statement.* The study was conducted according to Good Clinical Practice guidelines, the Declaration of Helsinki, US 21CFR Part 50—Protection of Human Subjects, and Part 56—Institutional Review Boards, and pursuant to state and federal HIPAA regulations (adni.loni.usc.edu). Study subjects (Table S1) and/or authorized representatives gave written informed consent at the time of enrollment for sample collection and completed questionnaires approved by each participating site Institutional Review Board (IRB). The authors obtained approval from the ADNI Data Sharing and Publications Committee for data use and publication, see documents http://adni.loni.usc.edu/wp-content/uploads/how_to_apply/ADNI_Data_Use_Agreement.pdf and http://adni.loni.usc.edu/wp-content/uploads/how_to_apply/ADNI_Manuscript_Citations.pdf, respectively.

*Example Dataset 1.* Seven hundred and thirty-six individual post mortem tissue samples from the dorsolateral prefrontal cortex BA9 of LOAD patients ($N = 376$), HD patients ($N = 184$), and nondemented subjects ($N = 173$) were collected and analyzed[18]. All autopsied brains were collected by the Harvard Brain Tissue Resource Center (HBTRC; GEO accession number GSE44772), and include subjects for whom both the donor and the next of kin had completed the HBTRC informed consent (http://www.brainbank.mclean.org/). Correspondingly, tissue collection and the research were conducted according to the HBTRC guidelines (http://www.brainbank.mclean.org/). Post mortem interval (PMI) was $17.8 \pm 8.3$ h, sample pH was $6.4 \pm 0.3$, and RNA integrity number (RIN) was $6.8 \pm 0.8$ for the average sample in the overall cohort. As described in ref. [18], RNA preparation and array hybridizations applied custom microarrays manufactured by Agilent Technologies consisting of 4720 control probes and 39,579 probes targeting transcripts representing 25,242 known and 14,337 predicted genes. Arrays were quantified on the basis of spot intensity relative to background, adjusted for experimental variation between arrays using average intensity over multiple channels, and fitted to an error model to determine significance[45]. Braak stage, general and regional atrophy, gray and white matter atrophy, and ventricular enlargement were assessed and cataloged by pathologists at McLean Hospital (Belmont, MA, USA). In addition, the severity of pathology in the HD brains was determined using the Vonsattel grading system[46].

*Example Dataset 2.* This study used a total of 911 individual data from the Alzheimer's Disease Neuroimaging Initiative (ADNI) (adni.loni.usc.edu). The participants underwent multimodal brain imaging evaluations, including amyloid-PET, tau-PET, and/or structural MRI. The ADNI was launched in 2003 as a public–private partnership, led by Principal Investigator Michael W. Weiner, MD. The primary goal of ADNI has been to test whether serial MRI, PET, other biological markers, and clinical and neuropsychological assessments can be combined to measure the progression of mild cognitive impairment (MCI) and early AD.

In a subset of 744 participants, the Affymetrix Human Genome U219 Array (www.affymetrix.com) was used for GE profiling from blood samples. Peripheral blood samples were collected using PAXgene tubes for RNA analysis[47]. The quality-controlled GE data includes activity levels for 49,293 transcripts. Molecular PET and MRI images quantifying seven different biological properties were mapped in vivo using the following techniques: structural MRI (for structural tissular properties; $N = 911$), fluorodeoxyglucose PET (for glucose metabolism; $N = 799$), Florbetapir PET (for Aβ deposition; $N = 906$), Arterial Spin Labeling (ASL, for cerebral blood flow; $N = 341$), resting functional MRI (for neuronal activity at rest; $N = 186$), [18]F-AV-1451 PET (for tau deposition; $N = 266$), and diffusion weighted MRI (for structural brain connectivity; $N = 128$). The preprocessing of the imaging data have been previously described in ref. [15]. For the first six mentioned imaging modalities, representative regional values were calculated for 78 regions covering all the gray matter[48]. The diffusion weighted MRI data were employed for whole-brain region–region structural connectivity (connectome) mapping. All the participants were also characterized cognitively using the mini-mental state examination (MMSE), a composite score of executive function (EF), a composite score of memory integrity (MEM)[49], and Alzheimer's Disease Assessment Scale-Cognitive Subscales 11 and 13 (ADAS-11 and ADAS-13, respectively). Also, they were clinically diagnosed at baseline as healthy control (HC), early mild cognitive impairment, late mild cognitive impairment, or probable Alzheimer's disease patient (LOAD).

*Example Dataset 3.* Finally, we simulated three datasets (see data and demo script included at the *NeuroPM-box*'s downloading page). Data were generated using a previously validated

method[10], with Matlab codes are available at https://github.com/ucl-mig/SuStaInMatlab). We set the number of subtypes to be three (results in Fig. S1A–C, with two "diseased" subpopulations and a control subgroup) and four (results in Fig. S1D–F, with three "diseased" subpopulations and a control subgroup). The number of subjects for each case was set to be 500 (results in Fig. S1A–F) or 1000 subjects (results included only in the demo). To each dataset, the number of informative biomarkers was set to be 50, adding other 50 randomly distributed biomarkers, for a total of 100 features (i.e. 50 informative, 50 non-informative). For each dataset, the subjects' stages were simulated using a uniform distribution, and the progression pattern of each subtype was defined according to a linear z-score model, parameterized by a sequence of z-score events with a random monotonic ordering (see original publication[10]).

See Table S1 for the corresponding demographic characteristics (Datasets 1 and 2).

## Methods

**cTI definition.** The inference of contrasted pseudotemporal trajectories (and characterizing disease progression and heterogeneity) consists of five main steps[16]:

(i) Optional data adjustment for confounding factors. This step is strongly recommended for experimental data in which different conditions (e.g. technical procedures) may affect the quantitative comparison of observations and subsequent identification of relevant biological components. For instance, in this study before applying the cTI approach, each gene transcript's activity was adjusted for relevant covariates using robust additive linear models with pair-wise interactions[50]. Specifically, Dataset 1 GE and histopathological data (HBTRC) were adjusted for PMI in hours, age, gender, and educational level. Dataset 2 GE (ADNI) was controlled for RIN, Plate Number, age, gender, and educational level.

(ii) Optional initial selection of features most likely to be involved in a trajectory across the entire population (recommended for high-dimensional data). The unsupervised method proposed by Welch et al.[40] is applied, scoring features by comparing sample variance and "neighborhood variance".

(iii) Data exploration and visualization via contrastive Principal Component Analysis (cPCA[21]). This technique identifies low-dimensional patterns that are enriched in a target dataset (e.g. a diseased population) relative to a comparison background dataset (e.g. demographically matched healthy subjects). By controlling the effects of characteristic patterns in the background, cPCA allows visualizing specific data structures missed by standard data exploration and visualization methods (e.g. traditional PCA, Kernel PCA). Specifically, if $C_{target}$ and $C_{background}$ are the covariance matrices of the target and background dataset, the directions returned by cPCA are the singular vectors of the weighted difference of the covariance matrices: $C_{target} - \alpha C_{background}$. The contrast parameter $\alpha$ represents the trade-off between having the high target variance and the low background variance. Multiple values of $\alpha$ are used (i.e. 100 logarithmically equally spaced points between $10^{-2}$ and $10^2$). Instead of choosing a single $\alpha$, the resulting subspaces for all the $\alpha$-values are clustered (based in their proximity in terms of the principal angle and spectral clustering[51,52]) in a few subspaces. The data are then projected onto each of these few subspaces, revealing different trends within the target data. While the original cPCA algorithm proposes to select the final subspace via visual examination, we chose automatically the subspace that maximize the clustering tendency in the projected target data, relative to the clustering tendency in the background population.

(iv) Individual pseudo-time calculation and subtyping according to the proximity to the background population in the contrasted principal components space (cPC). For this, we first calculate the Euclidean Distance Matrix among all the subjects and the associated minimum spanning tree (MST). The MST is then used to calculate the shortest trajectory/path from any subject to the background subjects. Each specific trajectory consists of the concatenation of relatively similar subjects, with a given behavior in the data's dimensionally reduced space. The position of each subject in his/her corresponding shortest trajectory reflects the individual proximity to the pathology-free state (the background) and, if analyzed in the inverse direction, to advanced disease state. Thus, to quantify the distance to these two extremes (background or disease), an individual pseudo-time score is calculated as the shortest distance value to the background's centroid, relative to the maximum population value (i.e. values are standardized between 0 and 1). Finally, spectral clustering[52] is performed over the cPC-based Euclidean Distance Matrix to identify subjects subtrajectories in the contrasted space. Note that, due to similar probabilities, some subjects may be assigned to multiple subtrajectories, thereby implying that the subtrajectories may overlap. Assignment to multiple subtrajectories is

particularly possible in the early stages of a disease, either due to the algorithm being unable to distinguish between different disease paths, or due to real biological effects (e.g., two disease variants with a common or similar starting process).

(v) Estimation of features relevance/influence. The total contribution $C_i$ of each data feature $i$ to the obtained reduced representation space (and the pseudotemporal trajectories) is quantified as[53]

$$C_i = 100 \sum_{j=1}^{N_{cPC}} \left( \lambda_j^{norm} \frac{\omega_{i,j}^2}{\sum_{k=1}^{N_{genes}} \omega_{i,j}^2} \right), \tag{1}$$

where $\lambda_j^{norm} = (\lambda_j - \min\lambda)/\sum_{k=1}^{N_{total}}(\lambda_k - \min\lambda)$ is the normalized eigenvalue of the contrasted principal component $j$, $\min\lambda$ is the minimum obtained eigenvalue, $N_{total}$ is the original number of contrasted principal components, $N_{cPC}$ is the number of contrasted principal components with $\lambda_j^{norm}$ over a predefined cut-off value (i.e. 0.025), $\omega_{i,j}$ is the loading/weight of the feature $i$ on the component $j$, and $N_{features}$ is the total number of features considered in the dimensionality reduction analysis.

**ESM definition.** The brain is modeled as a system with $N_{rois}$ structurally inter-connected gray matter regions[12] covering the brain's whole gray matter (in this study, $N_{rois} = 78$ based on a known anatomical brain parcellation[48]), where each region $i$ ($i = 1..N_{rois}$) is characterized by its temporal value ($S_i$) of infection-like "agent" (inf-A) accumulation (e.g. misfolded protein [MP] burden). Notice that $S$ can also be defined in probabilistic terms (see *NeuroPM-box* tutorial). The dynamic of this system, in terms of inf-A propagation and accumulation, will depend on the interactions between the inf-A "infected" and "non-infected" regions, with temporal changes in the regional $S_i$ values described by the non-linear differential equation:

$$\frac{dS_i}{dt} = \left(1 - S_i(t)/\max S\right)\varepsilon_i(t) - \delta_i(t)S_i(t) \tag{2}$$

The first term on the right side of Eq. (2) represents the regional likelihood (or probability) of receiving infectious-like "agents" ($\varepsilon_i(t)$) if region $i$ is "non-infected", where $\max S$ is the maximum possible $S$ value (defined across all the population). The second term corresponds to the likelihood (or probability) of being clean of inf-A at time $t$ ($\delta_i(t)$) if region $i$ was "infected" before. To consider the fact that a particular "infected" sub-region in brain region $i$ can potentially "infect" neighboring sub-regions, $\varepsilon_i(t)$ is modeled as the accumulation of exogenous and endogenous infectious-like factors:

$$\varepsilon_i(t) = \sum_{j \neq i} Pa_{j \to i}\beta_j^{ext}(t)S_j(t) + Pa_{i \to i}\beta_i^{int}(t)S_i(t), \tag{3}$$

where $Pa_{j \to i}$ is the weighted anatomical connection value between the regions $j$ and $i$, $\beta_j^{ext}(t)$ is the extrinsic "infection" rate of region $j$, and $\beta_i^{int}(t)$ is the intrinsic "infection" rate. We assume that inf-A diffuse from regions of higher concentration to regions of lower concentration. With a high inequality in the inf-A accumulation levels of all the regions causing an increase in the extrinsic propagation across the entire brain, and a decrease in the intrinsic fraction that stays in each seed region. These effects are modeled as

$$\beta_i^{ext}(t) = g(t)\beta_i(t), \tag{4}$$

$$\beta_i^{int}(t) = (1 - g(t))\beta_i(t),$$

where $g(t)$ is a global tuning variable (*Gini* coefficient[54]) that quantifies the temporal inf-A accumulation inequality among the different brain regions, and $\beta_i = \beta_i^{ext} + \beta_i^{int}$ is the total "infection" rate of the region $i$. A value of 0 for $g$ reflects perfect equality across all regions, and a value of 1 corresponds to a complete inequality.

In the *NeuroPM-box*, clearance and production rates can be modeled as constant values ($\beta_o$, $\delta_o$) across all regions or as sigmoid functions depending on the local $S$ values. When opting for a sigmoid relationship, $\beta_i$ is defined as

$$\beta_i(t) = \beta_i(P_i, \beta_o) = 1 - e^{-\beta_o S_i(t)/\max S}, \tag{5}$$

being $\beta_o \in [0, +\infty]$ an unknown constant parameter. Notice that a high inf-A accumulation at region $i$ will imply a high probability of producing new infectious-like factors. Similarly to $\beta_i(t)$, the regional clearance rate after "infection" ($\delta_i(t)$) is expressed as a function of $S_i(t)$ and a constant parameter. However, the regional capacity to clear/remove infectious-like agents will decrease with the increase in inf-A accumulation, following a decreasing exponential relationship:

$$\delta_i(t) = \delta_i(P_i, \delta_o) = e^{-\delta_o S_i(t)/\max S}, \tag{6}$$

where $\delta_o \in [0, +\infty]$ is also an unknown constant parameter. We hypothesize that $\beta_o$ and $\delta_o$ will depend on the specific MP under study, as well as on the individual

characteristics (e.g. genetic properties, lifestyle, environmental conditions). In sum, the ESM approach depends on two main unknown parameters $(\beta_o, \delta_o)$, which control the continuous competition between the infectious-like agents and the system's clearance response.

For optimizing the model, each participant's multimodal and longitudinal data are used to identify the ESM's fundamental Eq. (2). The following optimization function is used in combination with MATLAB's *MultiStart* algorithm[23]:

$$\mathscr{L}(\varnothing) = \sum_{k=1}^{N_t} \sum_{i=1}^{N_{rois}} \left( S_i(t_k) - \hat{S}_i(t_k; \varnothing) \right)^2, \quad (7)$$

where $N_t$ is the number of available longitudinal time points for the participant, and $\hat{S}_i(t_k; \varnothing)$ is the corresponding estimated value at the time point $t_k$ for the set of model parameters $\varnothing$.

**MCM definition**. The brain is modeled as a dynamic multifactorial causal system[14], where (i) each system node models a relevant biological factor (quantified via an imaging modality) at a given brain region and (ii) alterations in each biological factor are caused by direct factor–factor interactions and/or external inputs. For example, in the presented results (MCM subsection), we considered $N_{factors} = 6$ different biological factors (i.e. vascular flow, Aβ deposition, tau deposition, glucose metabolism, functional activity at rest, and gray matter density), each measured at $N_{rois} = 78$ brain gray matter regions[48]. Each node, corresponding to a given biological factor $m$ and region $i$, is characterized by its alteration/disequilibrium level, $S_i^m \in \mathbb{R}$, reflecting the distance to an initial baseline state ($S_i^m = 0$, $S_i^m < 0$, or $S_i^m > 0$ for non-alteration, decrease, or increment, respectively). In general, this system is defined by the [$N_{factors} N_{rois} \times 1$] state space vector $S(t) = \left[ S_1^1(t), S_2^1(t), \dots, S_i^m(t), \dots, S_{Nrois}^{N_{factors}}(t) \right]^T$ and $A(t)$, the *dynamic multifactorial direct interaction network*, where each directed edge corresponds to a factor–factor or a region–region interaction.

The dynamic behavior of the proposed brain system will depend on (i) the local direct interactions among all the biological factors, constrained within each brain region, (ii) the potential propagation of factor-specific alterations through "physical" networks (i.e. anatomical and/or vascular networks), and (iii) the influence of external inputs. These processes can be described by the differential equation model:

$$\begin{cases} \frac{dS}{dt} = A(t)S + Bu, \\ \text{Cog} = \beta S, \end{cases} \quad (8)$$

where $A(t)$ is a [$N_{factors} N_{rois} \times N_{factors} N_{rois}$] asymmetric network/matrix characterizing all the multifactorial interactions at time $t$. It depends on model parameters that are estimated during model fitting, and on the brain's connection properties, estimated a priori. $B$ is an [$N_{factors} N_{rois} \times M$] input matrix ($M \leq N_{factors} N_{rois}$) that identifies $M$ nodes (brain regions of any specific biological factor or factors) controlled by an outside controller[55,56]. $u(t) = [u_1(t) \dots u_M(t)]$ is the associated time-dependent input signal. Cog is a cognitive variable of interest modeled by additive linear relationships, considering the brain's multifactorial alterations as modulators (with weights defined by the vector $\beta$, estimated a priori at the population level).

The optimum input signal to control the described brain system can be estimated as[57]

$$u_{opt}^{S_0 \rightarrow S_f}(t) = -B^T e^{A^T(t_f - t)} W^{-1} D, \quad (9)$$

where $W$ is the controllability gramian matrix[58]:

$$W(t_0, t_f) = \int_{t_0}^{t_f} e^{A(t_0 - t)} BB^T e^{A^T(t_0 - t)} dt. \quad (10)$$

$D = \left( e^{A(t_f - t)} S(t_0) - S(t_f) \right)$ is the difference between the final and desired final state space vector under the free and controlled evolution, respectively. Finally, the cost-energy function associated to the set of nodes $B$ with optimum strategy $u_{opt}^{S_0 \rightarrow S_f}$ is calculated as[14]

$$J(B, u_{opt}) = \int_{t_0}^{t_f} \left( u_{opt}(t) \right)^T u_{opt}(t) dt. \quad (11)$$

For optimizing the model, each participant's multimodal and longitudinal data is used to identify the MCM's fundamental Eq. (8), in the absence of external signals (i.e. $u(t) = 0$). For each participant, the following optimization function is used in combination with MATLAB's *MultiStart* algorithm[23]:

$$\mathscr{L}(\varnothing) = \sum_{k=1}^{N_t} \sum_{m=1}^{N_{factors}} \sum_{i=1}^{N_{rois}} \left( S_i^m(t_k) - \hat{S}_i^m(t_k; \varnothing) \right)^2, \quad (12)$$

where $N_t$ is the number of available longitudinal time points for the participant, $S_i^m(t_k)$ is the observed alteration level for factor $m$ and brain region $i$, at the time point $t_k$, and $\hat{S}_i^m(t_k; \varnothing)$ is the corresponding estimated value for the set of model parameters $\varnothing$.

**pTIF definition**. To evaluate at the individual level the effectiveness of all possible one-target or combinatorial therapies, for each biological factor or combination of factors, expressions (9–11) are used to estimate the optimum input signal and associated cost-energy for stopping each patient's brain deterioration (i.e. keeping the patient's brain properties at a stationary state) and also for conducing each patient from its current state to a typically healthy state (i.e. the mean pattern observed for HC subjects). For a single-target intervention (i.e. based on a unique driving biological factor) the input matrix $B$ (Eq. 8) is constructed with one for all the nodes/regions corresponding to this factor, and zero for all the other nodes/regions. Similarly, for a combinatorial-target intervention, the matrix $B$ contains one for all the nodes/regions associated with the selected driving factors and zero for the rest[15]. Finally, for each subject, the individual pTIF is defined as the numeric multivariate vector with the estimated factor(s)-specific cost-energy values for all possible tested interventions (i.e. with a unique energy/deformation value for each hypothetical single target or combinatorial intervention). Note that, for $N_{factors} = 6$, the number of all possible single target or combinatorial interventions (up to a maximum of 6 factors) is 63.

**Statistics and reproducibility**. Three different data populations were used (total $N > 4547$), including post mortem ($N = 736$; see "Example Dataset 1" in the section "Data") and in vivo ($N = 911$; "Example Dataset 2") neurodegenerative individuals, and synthetic subjects ($N > 2900$; "Example Dataset 3"). We performed five independent analysis on different datasets to validate the reproducibility of the contrastive Trajectories Inference (cTI) method, obtaining highly consistent results for the different scenarios/data. The datasets included HBTRC GE, HBTRC histopathology, and three generated synthetic datasets. Similarly, ESM and MCM were extensively tested with the in vivo data from ADNI (see corresponding "Results" subsections). For the population-based cTI analysis, the GE and histopathological data from HBTRC were adjusted for PMI in hours, age, gender, and educational level. GE from ADNI was controlled for RIN, plate number, age, gender, and educational level. Neuroimaging data from ADNI were not adjusted, because the corresponding analyses were performed at the individual level. Traditional blinding was not relevant to our study. All our analyses were unsupervised, i.e. not requiring any a priori training/fitting on cognitive and/or behavioral variables.

**Using NeuroPM-Box in practice. Installation (timing 5–10 min)**: To run *NeuroPM-box* on Windows, Linux (or OS X), or *macOS* systems, you will need MATLAB's f Runtime 2019b. Please ensure the Runtime version corresponds to the MATLAB version used by NeuroPM-box (i.e. 2019b). MATLAB Runtime can be downloaded for free from https://www.mathworks.com/help/mps/qs/download-and-install-the-matlab-compiler-runtime-mcr.html.

(a) Download the software from https://www.neuropm-lab.com/neuropm-box.html

(b) *For Windows*: Run the provided *NeuroPM_box_installer.exe* file to install the software.

(c) *For Linux*: Call the startup script with the path to MATLAB or the Runtime root folder as an argument:
MATLAB installed:./run_NeuroPM_box.sh /*matlabroot*/matlab19b
Runtime installed:./run_NeuroPM_box.sh /*mcrroot*/matlab19b_runtime/v94
The root folder can be found in MATLAB by checking the variable "matlabroot". Also, in some cases, depending on your system's configuration, before installing you may need to provide appropriate permissions:
chmod u+x NeuroPM_box
chmod u+x run_NeuroPM_box.sh

(d) *For macOS*: Run the provided *NeuroPM_box_installer.app* file to install the software.

**Executing cTI algorithm (timing 5–15 min)**:
*Input data should be*:

(a) a .txt file, with row values representing observations and columns representing features.

(b) an ESM file. For each subject, a pseudo-time value (and trajectory position) will be calculated based on her/his baseline data (the unifactorial data; e.g. amyloid, tau). Regional values at baseline will be considered the features on the cTI analysis.

(c) a MCM file. Unless specified, all the biological factors (imaging modalities) available will be considered on the cTI (the user can specify if, by the contrary, would like to focus on only one factor/modality).

*Dimensionality reduction method*: in cTI, data exploration and visualization are performed via cPCA[21]. By controlling the effects of characteristic patterns in the background (e.g. pathology free and spurious associations, noise), cPCA and its non-linear version cKernel PCA[21] allow visualizing specific data structures missed by popular data exploration and visualization methods (e.g. PCA, Kernel PCA, t-SNE, UMAP). The user can select between "cPCA", "cKernel PCA", or "cPCA after applying a smoothing Kernel" to the data (which may reduce the influence of outliers). Of note, before the contrasted dimensionality reduction, all the data features will be 'boxcox' transformed (see https://www.ime.usp.br/~abe/lista/

pdfQWaCMboK68.pdf), centered to have mean 0 and scaled to have standard deviation 1.

*Features preselection*: For high-dimensional datasets (e.g. considerably more features than observations), it is necessary to perform an initial selection of features most likely to be involved in a trajectory across the entire population. By default, we apply the unsupervised method proposed by[40], which does not require prior knowledge of features involved in the process. Features are scored by comparing sample variance and neighborhood variance. A threshold is applied to select those features with higher score, e.g. you can keep the features with at least a 0.95 probability of being involved in a trajectory (i.e. around 5% of your features dimensionality). Please select the fraction of features that should be used by the cTI algorithm.

*Background population*: The cTI algorithm detects enriched patterns in the population of interest while adjusting by confounding components in the background population (i.e. subjects free of the main effect of interest). To define the background population, the user should provide the list of corresponding IDs, which can be entered by just copying the IDs in the interface, or by loading a ".txt" file in which each row have an ID. Importantly, if the cTI-data is entered as a ".txt" file, we will take as ID the subject's position on the data (e.g. subject number 10 in the data, at the row 10 of the data matrix, will have ID = 10). If, by the contrary, the cTI-data is coming from ESM/MCM data structures, the Background IDs must correspond with names of the individual folders in the main ESM/MCM folder.

*Target population (optional)*: By default, all the other subjects not defined as background are taken as the target population. However, the user may be interested in to defining the target with a particular subset of subjects (e.g. individuals notably advanced in a disease process). The algorithm will only use then the defined background and target to estimate the model parameters, while the corresponding transformations will be still applied to all the subjects in the data. To define the optional target population, the user should provide the list of corresponding IDs (following same format that for background population), which can be entered by just copying the IDs in the interface, or by loading a ".txt" file in which each row have an ID. Of note, this option is not valid when using the "cKernel PCA" method.

Δ **CRITICAL STEP**. Background and target populations have a strong influence on the cTI method[21]. We recommend defining these taking in to account the biological process of interest. For instance, when studying a given neurological disorder, ensure that the background subjects are free of the studied pathology and with similar demographic characteristics that the target population. The target may be constituted by an heterogenous population, but, if a subset of subjects with highly similar pathological stages/variants is considerably more abundant than subjects at other stages/variants, this subset could statistically dominate (and bias) the contrasted dimensionality reduction technique. In such cases, we recommend pre-defining the target with an equilibrated compendium of disease stages/variants.

*Adjusting by Covariables (optional)*: covariates can be included for linear data adjustment before the feature preselection (if selected) and the trajectories inference analysis. The covariables should be entered as a.txt file, where rows correspond to observations in the main data, the first column to subjects/ observations IDs, and the other columns to different covariables. Of note: If a covariable has less than seven unique values, it will be considered as categorical, and it will be divided in an equivalent number of variables (e.g. for gender information, where one input variable typically has two unique values [female or male], we will replace this variable by two predictor variables, i.e. one for each gender).

*Additional parameters*: The user can define the maximum number of features obtained from the contrasted dimensionality reduction algorithm. Also, the algorithm allows to identify subsets of subjects following potentially different contrasted subtrajectories. The user can define the maximum number of possible subtrajectories.

*cTI Outputs* (saved in the input data's folder):

(a) 'cTI_IDs_pseudotimes_pseudopaths_'data_name'.txt': file containing the main cTI outputs. First column corresponds to subjects' IDs. Second column to individual pseudo-time values (a value per subject). From the third to the last column (as many columns as different contrasted subtrajectories identified), the sub-trajectory or subtrajectories to which each subject belongs to. As mentioned, each subject can belong to more than one sub-trajectory (e.g. when two disease variants overlap at their beginning). In such cases, individual subtrajectories are sorted from maximum to minimum probability.

(b) 'cTI_cPCs_' data_name '.txt': obtained contrasted principal components.

(c) 'cTI_weights_' data_name '.txt': loadings/weights (one column per initial feature).

(d) 'cTI_features_contributions_' data_name '.txt': total features contribution on the final contrasted space (a value for each feature included on the analysis).

(e) 'cTI_features_preselected_' data_name '.txt': When feature preselection is performed, this file contains the indices of those features most likely to be involved in a trajectory across the entire population, which is subsequently used in the cTI analysis. Of note, if the features are preselected, the saved feature loadings/weights and total contribution values will correspond only to the preselected features.

**Executing ESM (timing 5–25 min per subject):**

*Organizing your data for ESM*: The software provides an automatic tool to import all the needed data for ESM evaluation (on the main interface, click "Complementary_tools"). The data should be organized individually, with a folder per subject. For compatibility across different models in the toolbox, the images can be organized in the same way that for the multimodal models (e.g. MCM). In case the software detects multiple imaging modalities, the user will be asked about which one should be used for ESM. Each subject's folder should include (see Fig. 6):

(a) Brain images (.nii or.mnc) corresponding to each biological factor of interest, e.g:
*factor_1_t0.nii, factor_1_t1.nii, factor_1_t2.nii, factor_1_t3.nii*
*factor_2_t0.nii, factor_2_t1.nii, factor_2_t2.nii,*
*factor_3_t0.nii, factor_3_t1.nii, factor_3_t2.nii, factor_3_t3.nii...*
Include as many factors and time points as available. Of note, t1, t2, t3... should be numeric values.

(b) Gray matter parcellations images (.nii or.mnc), e.g.:
*GM_parcellation_t0.nii, GM_parcellation_t1.nii, GM_parcellation_t2.nii, GM_parcellation_t3.nii...*
Importantly, if there is only a common population parcellation at the group level (e.g. coming from other study/template), the parcellation files can be, alternatively, saved at the root folder containing all the subject's folders. In any case, please be sure to include at least one parcellation image for each subject or for the whole population.

(c) Connectomes (mandatory, e.g. anatomical and/or vascular networks) files: Multiple (.txt or.csv) files (one for each available time point, with rows and columns corresponding to brain regions), for example:

'*connectome1_t0.txt', 'connectome1_t1.txt', 'connectome1_t2.txt'...*
Importantly, if the connectivity information is only available at the group level (e.g. coming from other study/template), the connectome files can be, alternatively, saved at the root folder containing all the subjects' folders. In any case, please be sure to include at least one connectome matrix, with same number of rows and columns (i.e. regions) that your parcellation.

*Data standardization (optional)*: The voxel or regional values can be standardized at the individual level using user-defined reference regions (e.g. cerebellum). When using this option, the user should define:

(a) labels of reference regions, corresponding with the regions' numeric values in the provided parcellation images.

(b) if the reference regions should NOT be removed for the posterior modeling analysis (by default, the references are removed).

*Probabilistic data standardization (optional)*: the original ESM version was defined in probabilistic terms. Although the current implementation can work with raw data values (default), the user can opt to convert the raw voxel values to probabilities, by comparing each voxel with the distribution of maximum in the reference regions.

*Reversing scale (optional)*: by default, it is considered that higher signal values in the images will be more reflective of the process of interest (e.g. misfolded proteins deposition quantified with PET imaging). However, for some data modalities (e.g. gray matter density), a lower value would imply a stronger effect (e.g. structural atrophy). For those cases, the user can opt to reverse the data's scale via a linear scaling (the numerical scale will runs in the opposite direction).

*Input file for ESM optimization*: After using "Organizing input for ESM" in "Complementary Tools", input the file named "Input_data_..._ESM.mat", which should be saved in the folder "ESM_data_results" inside your data's initial directory. All optimization results will be also saved in this file as well as in text files.

*The ESM can be optimized at*:

(a) the individual level (only if longitudinal data are available). A minimum of three time points is required (subjects without enough data will not be analyzed). Individual parameters (clearance, production, onset time) will be saved as a.txt file where rows are subjects and columns are IDs, effective clearance, effective production and onset time (if required). Additionally, for visualization purposes, these variables will be saved in the original "_ESM. mat" file.

(b) the population level. Δ **CRITICAL STEP** First, use the "contrastive Trajectories Inference (cTI)" algorithm, where a pseudo-time value will be obtained for each subject based on his/her baseline data (see instructions about). Alternatively, use your own subjects' stratification, coming from using cTI on a different data modality (e.g. molecular, clinical) or from a different computational method (for using this option, see "Adding new stratification to ESM/MCM files" in "Complementary Tools"). Then, the subjects will be ordered according to their characteristic pseudo-path and pseudo-time, constituting a pseudo-longitudinal data for ESM optimization. Subgroup parameters (clearance, production, onset time) will be saved in a. txt file, where rows will be subgroups (each corresponding to subjects belonging to a characteristic pseudo-path), and columns will have: subject(s) ID, effective clearance, effective production and onset time (if required). Additionally, for the visualization purposes, these variables will be saved in the original "_ESM.mat" file.

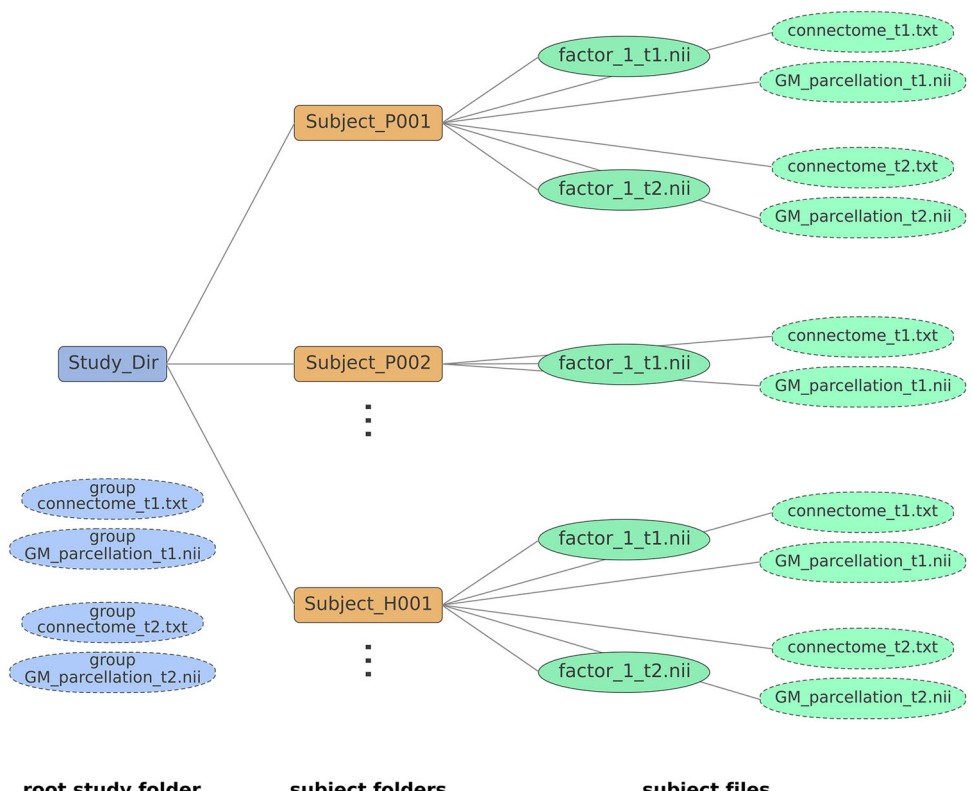

**root study folder subject folders subject files**

**Fig. 6 Schematic for data organization in an ESM study.** As mentioned, if the brain parcellation(s) and/or connectivity information are only available at the group level (e.g. coming from a template or another study), the parcellation(s) and/or connectome files can be, alternatively, saved at the root folder containing all the subjects. Please be sure to include at least one connectome matrix for the first modality, with same number of rows and columns (i.e. regions) that your parcellation.

*Production and clearance factors can be defined as follows:*

(a) Sigmoidal (default): following sigmoid functions that depend on global production, or clearance rates, and on the current regional value. In this case, each brain region has its own effective production or clearance rate, which may change with time. The effective regional production is assumed to increase with the local signal (e.g. the more amyloid a region has, the higher chance that it will produce and spread more amyloid seeds). Contrary, effective clearance is assumed to decrease with the local signal (i.e. more "infected" regions are less able to clean the accumulated "agents").

(b) Constants, with same rate across all the spreading process.

*Epicenters identification*: The spreading process under study is assumed to start in a set of specific brain regions, from which the "agent" propagates across physical brain connections. If the epicenter regions are known, they can be predefined by the user and the optimization algorithm will continue from these. Otherwise, the epicenter(s) will be estimated as the region[s] with the highest value[s] at the onset time. Both backward and forward integration procedures will be used for estimating the most likely epicenter(s), using the maximum number of allowed epicenter regions defined by the user.

**Δ CRITICAL STEP.** When working with some imaging modalities (e.g. SUVr PET), a non-zero regional value does not necessarily implies the presence of the studied "agent" (e.g. amyloid or tau deposition) but just background fluctuations on the image signal (determined, for example, by different physiological factors and/or random noise). Commonly, in literature, a positivity threshold is applied to detect "agent" presence or not. Consequently, for estimating the regional epicenters, our algorithm allows to predefine a maximum (non-zero) value below which the regions are still considered free of "agent" presence but with their typical background "noise". Only regions over this value will be considered as likely epicenters (and, correspondingly, all regions below are considered non-epicenters). The user may see some improvement in model fit when using a non-zero value, due to the non-epicenters will not be forced to be zero at the onset time (i.e., increasing correspondence with reality, in which regions may never have an exact zero value, depending on the imaging modality used).

Of note, increasing "too much" the maximum value for non-epicenters may result in data overfitting, because the non-epicenters may have a large enough range, and the corresponding numerical flexibility, to adapt to the studied process. That is, if the "agent" positivity threshold is not conservative enough (i.e. too high), all brain regions can potentially behave as epicenters. We suggest a value about or

below the 5% of the typical maximum value in the analyzed imaging modality (or about/below 0.05 if working with probabilistic values).

*Estimating onset time*: this measure should be interpreted as the time at which the intra-brain spreading process under study started (e.g. the age at which amyloid and tau proteins appeared and started propagating in the brain). If known, the user can provide the onset time. Otherwise, it will be estimated by the optimization algorithm. In such a case, the user would need to provide the minimum possible value (default: zero).

*ESM Outputs* (saved as.txt files in the folder "ESM_data_results" inside your data's initial directory, and as MATLAB variables in the ESM's.mat file):

(a) 'ESM_subject_…_ACCURACY.text': model accuracy (in %).

(b) 'ESM_subject_…_resnorm.text': 2-norm of the residuals.

(c) 'ESM_subject_…_parameters.text': obtained model parameters (production and clearance, respectively) in their original numeric scale.

(d) 'ESM_subject_…_effective_production.text': production and clearance model parameters may be difficult to interpret in the original numeric scale. To facilitate parameter comparability across subjects, the individual production parameter's marginalized across all possible "agent" concentration/probability values, obtaining the effective individual production rate.

(e) 'ESM_subject_…_effective_clearance.text': similarly, the individual clearance parameter's marginalized across all possible regional concentration/probability values, obtaining the effective individual clearance rate.

(f) 'ESM_subject_…_sorted_most_likely_epicenters.text': when the epicenter regions are not provided by the user, a list of most likely epicenters will be provided, based on the model's optimization.

(g) 'ESM_subject_…_S0.text': obtained "agent" concentration/probability values at the estimated onset time, i.e. initial perturbation triggering the spreading process. Notice that the solution is not necessarily sparse (i.e. all regions could have a non-zero value), but those regions with highest values (over a positivity threshold) should be considered the most likely propagation epicenters.

(h) 'ESM_subject_…_onset_time.text': estimated (or provided) time at which start the intra-brain "agent" spreading process.

(i) 'ESM_subject_…_simulated_data.text': by default, the optimized model parameters will be used to generate/simulate 30 data points, equally positioned in time from: the estimated (or provided) onset time, until the last available time point plus the half of the longitudinal time window of the

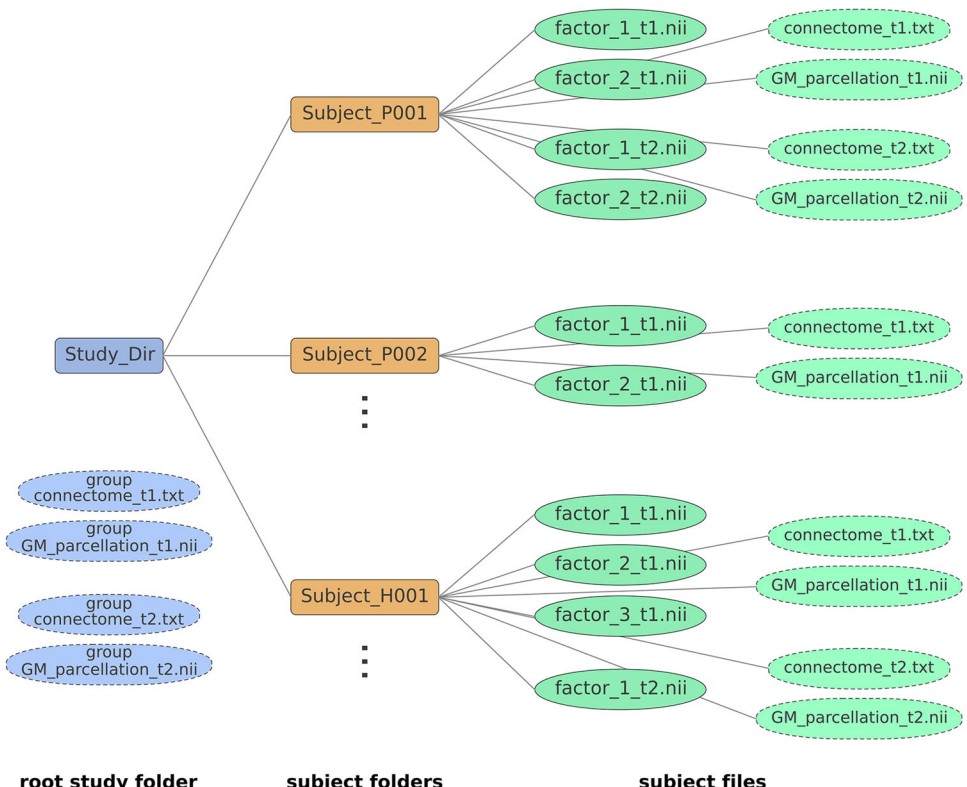

**root study folder**      **subject folders**      **subject files**

**Fig. 7 Schematic for data organization in an MCM study.** As mentioned above, if the brain parcellation(s) and/or connectivity information are only available at the group level (e.g. coming from a template or another study), the parcellation(s) and/or connectome files can be, alternatively, saved at the root folder containing all the subjects. Please be sure to include at least one connectome matrix for the first modality, with the same number of rows and columns (i.e. regions) that your parcellation.

subject's real data. In addition, the model will generate the data at the observed time points, simulating a total of $N_{times} = 30 +$ subject time points. This simulated data (saved here as a $[N_{regions}*N_{modalities} \times N_{times}]$ matrix) can be of particular interest to visualize spatiotemporal brain changes in a continues time scale. Also, the generated data at the observed time points can be used for model validation, comparing with the real observed data.

(j) 'ESM_subject_...._simulated_times.text': corresponding time values to the generated/simulated data.

For the case when the model is applied at the population level (after using the pseudo-times and subtrajectories from *cTI*), all the same model outputs will be saved for each previously identified sub-trajectory/subgroup, e.g. 'ESM_Subgroup_1_ACCURACY.text' and 'ESM_Subgroup_2_ACCURACY.text'.

**Executing MCM (timing 5–25 min per subject):**

*Organizing your data for MCM*: The software provides an automatic tool to import all the needed data for MCM evaluation (on the main interface, click "Complementary_tools"). The data should be organized individually, with a folder per subject, each subject's folder including (see Fig. 7):

(a) Brain images (.nii or.mnc) corresponding to each biological factor of interest, e.g.:
*factor_1_t0.nii, factor_1_t1.nii, factor_1_t2.nii, factor_1_t3.nii*
*factor_2_t0.nii, factor_2_t1.nii, factor_2_t2.nii,*
*factor_3_t0.nii, factor_3_t1.nii, factor_3_t2.nii, factor_3_t3.nii*
Include as many factors and time points as available. t1, t2, t3… should be numeric values.

(b) Gray matter parcellations images (.nii or.mnc):
*GM_parcellation_t0.nii, GM_parcellation_t1.nii, GM_parcellation_t2.nii, GM_parcellation_t3.nii…*
Importantly, if there is only a common parcellation at the group level (e.g. coming from another study/template), the parcellation files can be, alternatively, saved in the root folder containing all the subjects' folders. In any case, please be sure to include at least one parcellation image for each subject or for the whole population.

(c) Connectomes (mandatory, e.g. anatomical and/or vascular networks) files:
Multiple (.txt or.csv) files (one for each connectome modality and each time point, with rows and columns corresponding to brain regions), named for example:
*connectome1_t0.txt, connectome1_t1.txt, connectome1_t2.txt…*

Optionally, if a second connectome modality is available, name it as:
*connectome2_t0.txt, 'connectome2_t1.txt, connectome2_t2.txt…*
Importantly, if the connectivity information is only available at the group level (e.g. coming from other study/template), the connectome files can be, alternatively, saved at the root folder containing all the subjects' folders. In any case, please be sure to include at least one connectome matrix for the first modality, with same number of rows and columns (i.e. regions) that your parcellation.

(d) Output variables file (optional, e.g. cognitive/clinical evaluations):
In each subject's folder, include a file named 'outputs_variables.txt', with as many rows as time points, and organized as
*value_variable1 value_variable2 (…) value_variableN evaluation_t0*
*value_variable1 variable2 (…) value_variableN evaluation_t1*
Include as many time points as available. For each missing value, use a 'NaN'.

(e) External Inputs file (optional, e.g. drugs intake): In each subject's folder, include a file named 'input_variable.txt', with as many rows as time points or inputs presented, and organized as
*starting_time1 finishing_time1 value_input_intensity (constant value if not changing) specific_target_regions (numbers in GM parcellation, leave empty if all regions)*
*starting_time2 finishing_time2 value_input_intensity specific_target_regions*
*starting_time3 finishing_time3 value_input_intensity specific_target_regions*
Include as many time points as available. All entrances should be numeric values. *value_input_intensity* should be a constant value reflecting the intensity of the stimulus in the specified time window. *specific_target_regions* should be the numbers of the targeted regions in the GM parcellation, leave empty if all regions were targeted.

**Δ CRITICAL STEP**. When working with longitudinal imaging data, it is common to have subjects with missing time points and/or imaging modalities. During the data's organization, the user can opt to impute the missing data via the trimmed scores regression algorithm[42]. We strongly recommend using this option, particularly when the subjects may have different acquired time points and imaging modalities (e.g. ADNI data). Otherwise, be sure to have a complete dataset for each subject. If imputation is not selected, subjects with missing imaging modalities will be removed from the analysis.

*Input file for MCM*: an MCM file. After using "Organizing input for MCM" in "Complementary Tools", you should have a file for MCM evaluation/

optimization, termed "Input_data_..._MCM.mat", and saved in the folder "MCM_data_results" inside your initial data's directory. All optimization results will be also saved in this file as well as in text files.

The MCM can be optimized at:

(f) the individual level (only if longitudinal data is available). A minimum of three time points is required (subjects without enough data will not be analyzed). Individual parameters (e.g. factor-to-factor causal interactions) will be saved as a.txt file where rows are subjects and columns are subject ID and parameters. Additionally, for visualization purposes, these variables will be saved in the original "_MCM.mat" file.

(g) the population level. Δ CRITICAL STEP. First, use the "contrastive Trajectories Inference (cTI) algorithm, where a pseudo-time value will be obtained for each subject based on his/her baseline data[16]. Alternatively, enter your own subjects' stratification, coming from using cTI on a different data modality (e.g. molecular, clinical) or from a different computational method (for using this option, see "Adding new stratification to ESM/MCM files" in "Complementary Tools"). Then, the subjects will be ordered according to their characteristic pseudo-path and pseudo-time, constituting a pseudo-longitudinal data for MCM optimization. Subgroup parameters (e.g. factor-to-factor causal interactions) will be saved in a.txt file where rows are subgroups (each corresponding to subjects belonging to a characteristic pseudo-path) and columns are subject ID, and model parameters. Additionally, for visualization purposes, these variables will be saved in the original "_MCM.mat" file. Please see also the "contrastive Trajectories Inference (cTI)" interface.

*Estimating initial system perturbation before baseline*: Use this option if you consider that the subjects may have an underlying "alteration" process started before their first imaging evaluation (e.g. a long-term neurodegenerative process started years time before the baseline). Importantly, do you know the clinical state of the subjects before the first "alteration"? Does any of the subjects in the current population is "free" of relevant dynamic changes (at least until their first evaluation)? If so, those subjects can be considered "controls" and, by specifying their IDs in a.txt file, you can help the model to have an approximation of the data's typical distribution before the potential perturbation under study occurred. If this information is available, we strongly recommend loading their IDs (i.e. the corresponding folder names of those subjects). Otherwise, the first evaluation will be taken as reference for quantifying potential brain alterations/perturbations, for each subject.

*Estimating initial system perturbation after baseline*: If you have information on the time when an external event/input perturbed the brain system, we recommend including this information in the data's organization. For MCM optimization, use then the "Known event(s) or input(s)" option, which will consider the corresponding information. If not, select "Unknown event(s) or input(s)". If "Estimate S0" is also selected in the "Optimization" panel, the algorithm will estimate the initial system perturbation.

Δ CRITICAL STEP. By default, we use a spline-based optimization method to solve the system of differential equations[59,60]. This results in a very fast optimization, without needing to propose initial parameters. Alternatively, if this option is not active, we will use a conventional gradient-based optimization method (trust-region-reflective algorithm[43,61]), which is started at multiple seed points for avoiding local minimum solutions. The former method takes from seconds to a few minutes per subject, while the latter method results in a significantly larger computational time, in the order of hours per subject. In our data, both methods provided comparable results.

*Considering external inputs (optional)*: Select this option to estimate the effects of any known external input on the brain system. For this, you should have previously included information about the external input in the data's organization process (see details above). The optimization algorithm will estimate a global measure of the impact that the input has on each considered biological factor.

*Estimate S0 (optional)*: Use this option if you would like to obtain an estimate of the initial perturbation on the brain system (i.e. what may have caused the initial propagation of biological alterations on the system).

*Regularization (optional)*: If you are using the "trust-region-reflective" algorithm (and not the spline smoothing) a Tikhonov regularization will be used during the parameters' optimization. The regularization may significantly improve the parameters, its robustness and biological interpretability, although it will also require a considerably larger computational time.

*Parallel calculus (optional)*: When using the "trust-region-reflective" algorithm (and not the spline smoothing), use this option if you would like to use your PC's multiple cores during model optimization. It may result in a significant reduction of the computational time.

*Number of iterations (optional)*: In order to refine the output parameters, the MCM is optimized multiple times. The higher the number of iterations, the longer the computational time, but potentially better results.

*MCM Outputs* (saved as.txt files in the folder "MCM_data_results" inside your data's initial directory, and as MATLAB variables in the MCM's.mat file):

Of note, here we will refer to $N_{regions}$ and $N_{modalities}$ as the number of brain regions and imaging modalities considered, respectively. Each biological factor corresponds to a given imaging modality.

(a) 'MCM_subject_' subject_ID '_accuracy_resnorm.txt': obtained model accuracy (in %) and 2-norm of the residuals, respectively.

(b) 'MCM_subject_' subject_ID '_parameters.txt': obtained model parameters in their original numeric scale. All parameters are also saved as different outputs according their specific role and biological interpretation on the model (see descriptions below). First $N_{modalities}*N_{modalities}$ corresponds to direct factor-to-factor interactions (rows are *seeds*, columns are *targets*; see *Effective causality* below). Second $N_{modalities}$ elements correspond to factor-specific scaling/weighting values associated to the intra-brain spreading processes (a high value for factor $m$ suggesting an strong role of the intra-brain spreading process for this factor; however, for post hoc analysis, we recommend to use instead the *Effective spreading* output described below). Next $N_{modalities}$ elements reflect the fraction [0,1] of factor/modality specific alterations spreading through the first brain network specified (e.g. if the user provided both anatomical and vascular brain connectomes, these output parameters would be reflecting the factor-specific fraction of spreading via the anatomical network, while the difference with 1 would reflect the fraction of spreading by the vascular network). Finally, if external input information's specified, the last $N_{modalities}$ parameters will correspond to the global direct influence of the input signal on each factor/modality considered (see also 'intervention_effects' and 'relative_intervention_effects' outputs, described below).

(c) 'MCM_subject_' subject_ID '_Effective_causality.txt': relative direct factor-to-factor influences (effective causal effects). Square matrix of size $[N_{modalities} \times N_{modalities}]$, where the element $n,m$ corresponds to the relative direct effect of factor $n$ over $m$ ($n{\to}m$) while accounting for all other factors interactions and intra-brain spreading, that is, the percent of regional changes in factor $m$ that are caused by the direct influence of factor $n$. It is calculated as 100 multiplied by the sum of the direct effects of $n$ over $m$, across all brain regions, relative to the sum of direct effects of all the biological factors over $m$ (including itself) and spreading effects.

(d) 'MCM_subject_' subject_ID '_Effective_X_initial.txt': initial estimation of the relative factor-to-factor influences (effective causal effects), obtained before the model's optimization via a regression analysis and only used as a priori input for model estimation (not recommended to be used in post hoc analysis).

(e) 'MCM_subject_' subject_ID '_Effective_spreading.txt': relative spreading of considered factors, where the element $m$ reflects the percent of the spatiotemporal changes in factor $m$ that are presumably caused by its spreading through brain physical connections (and not by factor-to-factor interactions or external inputs).

(f) 'MCM_subject_' subject_ID '_Effective_incoming.txt': relative incoming influences for considered factors. Similar to the in-strength measure in a directed network, element $m$ reflects the percent of regional changes in factor $m$ that are caused by the direct influences of all the other biological factors, excluding self-effects. This measure allows the identification of the most vulnerable and influenced biological factors (reflected in imaging modalities) during a given brain process.

(g) 'MCM_subject_' subject_ID '_Effective_outgoing.txt': relative outgoing influences for considered factors. Similar to the out-strength measure in a directed network reflects the percent of regional changes in all the considered biological factors that are caused by the direct influence of a given biological factor $n$, excluding self-effects. This measure can be particularly useful to detect the most influential biological factors during a brain process.

(h) 'MCM_subject_' subject_ID '_A_networks_optimum.txt': identified *multifactorial causal network* (matrix A in ref. [14]), with parameters controlling regional multifactorial causal interactions and effects propagation through physical networks (e.g. axonal and vascular connectomes). Matrix of size $[N_{regions}*N_{modalities} \times N_{regions}*N_{modalities}]$.

(i) 'MCM_subject_' subject_ID '_initial_perturbation.txt': estimated (or provided by the user if external input information's specified) initial system perturbation, i.e. a vector of size $[N_{regions} \times N_{modalities}]$, where first $N_{regions}$ elements correspond to perturbations in imaging modality/factor 1, second $N_{regions}$ elements to perturbations in modality/factor 1, and so on, until.

(j) 'MCM_subject_' subject_ID '_perturbation_time.txt': estimated (or provided by the user if external input information's specified) time at which happens the initial system perturbation.

(k) 'MCM_subject_' subject_ID '_intervention_effects.txt': if external input information's specified, this output consist of $N_{modalities}$ parameters corresponding to the estimated global direct influence of the input signal on each factor/modality considered (see also 'relative_intervention_effects' below).

(l) 'MCM_subject_' subject_ID '_relative_intervention_effects.txt': same that 'intervention_effects', but after normalizing each factor $m$'s corresponding value by the sum of direct effects of all the biological factors over $m$ (including itself), spreading- and input effects.

(m) 'MCM_subject_' subject_ID '_est_data_before_perturbation.txt': vector with $N_{regions}*N_{modalities}$ elements corresponding to estimated multifactorial regional values at the time that the initial perturbation occurred.

(n) 'MCM_subject_' subject_ID '_simulated_data.txt': by default, the optimized model parameters will be used to generate/simulate 30 multifactorial data points, equally positioned in time from: the estimated (or provided) onset time, until the last available time point plus the half of the longitudinal time window of the subject's real data. In addition, the model will generate the

data at the observed time points, simulating a total of $N_{times} = 30 +$ subject time points. This simulated data (saved here as a $[N_{regions}*N_{modalities} \times N_{times}]$ matrix) can be of particular interest to visualize spatiotemporal brain changes in a continues time scale. Also, the generated data at the observed time points can be used for model validation, comparing with the real observed data.

(o) 'MCM_subject_' subject_ID '_simulated_times.txt': corresponding time values to the generated/simulated data.

For the case when the model is applied at the population level (after using the pseudo-times and subtrajectories from *cTI*), all the same model outputs will be saved for each previously identified sub-trajectory/subgroup, e.g. 'MCM_Subgroup_1_ACCURACY.text' and MCM_Subgroup_2_ACCURACY. text'.

**Executing pTIF (timing 5–25 min per subject):**
*Input data for pTIF estimation*: the MCM's.mat file after optimization. See "Organizing input for MCM" in "Complementary Tools", and *Executing MCM* subsection.

*The pTIF can be estimated when*:

(a) having individual longitudinal data. Firstly, the MCM approach should be optimized at the individual level, using the available longitudinal data. Then, each individual multifactorial causal network will be analyzed, depending on the selected options (see below) to provide an individual pTIF (a vector with the required energy deformations to move a subject's state at the time of her/his final evaluation to a desired state). The corresponding pTIF values (and the chosen options) will be saved as "…pTIF.txt" in the MCM data/ results folder.

(b) having cross-sectional data for a relatively large population. **Δ CRITICAL STEP** First, use the "contrastive Trajectories Inference (cTI)" algorithm with your MCM file. Then, the subjects will be ordered according to their characteristic pseudo-path and pseudo-time, constituting a pseudo-longitudinal data for MCM optimization. Once the (sub)population MCM is optimized, the group's multifactorial causal network will be analyzed, depending on the selected options (see below), to provide an individual pTIF (a vector with the required energy deformations to conduce a subject's state at the time of the evaluation to a desired state). The temporal analysis will be based on the pseudo-time scale. All the individual pTIF values (and the chosen options) will be saved as "…pTIF.txt" in the MCM data/results folder.
*Control brain factors or output variables*: The MCM can be used as in silico evaluator of external inputs, which can focus on obtaining a desired state following two different control strategies:

(a) *Full control*: focuses on controlling all considered brain factors and regions.
(b) *Output control*: focuses on controlling cognitive/behavioral states, without necessarily modifying all the studied brain properties, and focusing on a specific output variable (e.g. a given cognitive metric).

*Desired system state*: When estimating the optimum signal to conduce the brain system or its outputs from a current state (last time point) to a desired state, the user should specify the final desired state, opting to:

(a) keeping stable the observed alterations (desired state is equal to initial state).
(b) reducing the current state's alterations to a given percent (provide a value between 0 [reducing all alterations to zero level] and 100 [causing no change]).
(c) reducing the current state's alterations to the mean level of the control subjects, which is the reference for calculating the alterations (this analysis can only be performed if control subjects were defined before MCM optimization).

*Intervention duration*: Time window of the hypothetical brain intervention. **Δ CRITICAL STEP** It must be in the same scale that the reported time in the observed data. However, if using a pseudo-time metric instead, be sure to always enter a value between 0 and 1. In this case, it is important to consider that the interval 0 to 1 may be equivalent to the whole time that takes the studied process (e.g. a disease covering years of progression). We recommend using a conservative number, for example, if a disease under study usually takes 10 years to develop, and you would like to simulate a 1-year intervention, define an intervention duration equal to 1/10.

*Targeting combinations of factors*: Define the number of factors (imaging modalities) that are going to be targeted during the hypothetical external intervention. For combinatorial interventions, we recommend combinations of as many biological factors as available imaging modalities.

*pTIF outputs* (saved as.txt in the input data's folder and as variables in the MCM's.mat file). Here we will refer to $N_{subjects}$, $N_{regions}$ and $N_{modalities}$ as the number of subjects, brain regions, and imaging modalities considered, respectively. Each biological factor corresponds to a given imaging modality.

(a) 'Results_Input_data_(…)_MCM_OPTIONS_for_pTIF.txt': options selected by the user for pTIF estimation (e.g. control strategy, input duration, combinations of factors).
(b) 'Results_Input_data_(…)_MCM_pTIF.txt': global pTIF matrix of size $[N_{subjects} \times N_{pTIF}]$. The individual pTIF corresponds to a numeric

multivariate vector with dimensionality determined by all possible "tested" interventions. The number of pTIF elements ($N_{pTIF}$) depend on the number of imaging modalities/factors considered and the number of factors combined (see *Targeting combinations of factors* option above). For instance, for $N_{modalities} = 6$, the number of all possible single-target or combinatorial interventions (up to a maximum of 6 factors) is 63, which finally defines the individual pTIF vector.

(c) 'Results_Input_data_(…)_MCM_regional_pTIF.txt': regional pTIF matrix of size $[N_{subjects} \times N_{regions}*N_{pTIF}]$. For each "tested" intervention, the global pTIF value (reported in the described global pTIF matrix, above) corresponds to the sum of all the energy-cost regional values required to cause the desired change. Contrary, the regional pTIF matrix include all the regional values before adding them, which provides a detailed characterization of the estimated factor(s)-specific changes for each brain region.

**Reporting summary**. Further information on research design is available in the Nature Research Reporting Summary linked to this article.

## Data availability
All synthetic data used are available at the software's downloading page. All real data used are publicly available at the Gene Expression Omnibus (GEO; www.ncbi.nlm.nih. gov/geo, accession number GSE44772) and the Alzheimer's Disease Neuroimaging Initiative (ADNI; www.adni.loni.usc.edu). For reproducibility purposes, anonymized IDs of the studied ADNI subjects will be provided upon request, as well as detailed information of the imaging modalities and time points analyzed for each subject.

## Code availability
*NeuroPM-box* software and its PDF tutorial are freely available at neuropm-lab.com/ neuropm-box.html. User-friendly standalone applications for *Linux, macOS,* and *Windows* systems are provided (importantly, MATLAB license and/or programming expertise are not required). Additionally, a demo script for testing/evaluation is provided, including a user guide to reproduce all results presented in Fig. S1.

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

## Acknowledgements

We want to give a special thanks to Ms. Sheila Linden for her generous funding support to this project. We also thank Joanne Clark (former Executive Director of the *Ludmer Centre for NeuroInformatics & Mental Health* at McGill) for her valuable manuscript revisions. This project was also undertaken thanks in part to the following funding awards to Y.I.-M.: the Canada Research Chair tier-2, the *Fonds de la recherche en santé du Québec* (FRQS) Research Scholars Junior 1, the Weston Brain Institute Rapid Response AD program 2018, and the New Investigator start-up grant from McGill University's *Healthy Brains for Healthy Lives Initiative* (*Canada First Research Excellence Fund*). In addition, the used neuroinformatic infrastructure at the McConnell Brain Imaging Center—McGill (from where is distributed the software) has been possible with financial support of Health Canada, through the Canada Brain Research Fund, an innovative partnership between the Government of Canada (through Health Canada) and Brain Canada, and the Montreal Neurological Institute. Dataset-2 collection and sharing for this project was funded by the Alzheimer's Disease Neuroimaging Initiative (ADNI) database (adni.loni.usc.edu; National Institutes of Health Grant U01 AG024904) and DOD ADNI (Department of Defense award number W81XWH-12-2-0012). ADNI is funded by the National Institute on Aging, the National Institute of Biomedical Imaging and Bioengineering, and through generous contributions from the following: AbbVie, Alzheimer's Association; Alzheimer's Drug Discovery Foundation; Araclon Biotech; BioClinica, Inc.; Biogen; Bristol-Myers Squibb

Company; CereSpir, Inc.; Eisai Inc.; Elan Pharmaceuticals, Inc.; Eli Lilly and Company; EuroImmun; F. Hoffmann-La Roche Ltd and its affiliated company Genentech, Inc.; Fujirebio; GE Healthcare; IXICO Ltd; Janssen Alzheimer Immunotherapy Research & Development, LLC; Johnson & Johnson Pharmaceutical Research & Development LLC; Lumosity; Lundbeck; Merck & Co., Inc.; Meso Scale Diagnostics, LLC; NeuroRx Research; Neurotrack Technologies; Novartis Pharmaceuticals Corporation; Pfizer Inc.; Piramal Imaging; Servier; Takeda Pharmaceutical Company; and Transition Therapeutics. The Canadian Institutes of Health Research is providing funds to support ADNI clinical sites in Canada. Private sector contributions are facilitated by the Foundation for the National Institutes of Health (www.fnih.org). The grantee organization is the Northern California Institute for Research and Education, and the study is coordinated by the Alzheimer's Disease Cooperative Study at the University of California, San Diego. ADNI data are disseminated by the Laboratory for Neuro Imaging at the University of Southern California. The authors who made direct contribution to this study have been listed as authors in this article. Data used in preparation of this article were in part obtained from the ADNI database (adni.loni.usc.edu). As such, the ADNI investigators contributed to the design and implementation of ADNI and/or provided data but did not participate in the data analysis or writing of this report. A complete listing of ADNI investigators can be found at: http://adni.loni.usc.edu/wp-content/uploads/how_to_apply/ADNI_Acknowledgement_List.pdf

## Author contributions

Y.I.-M. conceived the software's main interfaces, implemented the analytical source codes (for cTI, ESM, MCM, and pTIF), preprocessed and analyzed the presented data, and wrote the manuscript and tutorial drafts. F.C. implemented the software's visualization interface NeuroPM-viewer. A.A., T.R.B., and L.S.-R. tested early software versions with real data, providing valuable feedback. A.K. assisted with software compilation to standalone applications for public release. Q.A. and Y.I.-M. prepared the software's webpage. All authors contributed to constructive discussions regarding manuscript preparation.

## Competing interests

The authors declare no competing interests.
