## [Peer Review File · Communications Biology]

Reviewers' comments:

Reviewer #1 (Remarks to the Author):

In their manuscript Iturria-Medina et al. describe NeuroPM-box, a software that handles and unifies multiple heterogeneous data related to brain research. The different tools (e.g. trajectory-based gene expression analysis, spreading models) have been previously published. Additions and improvements of the published methods are highlighted. The methods were tested on several large data sets including simulated and experimental neurodegenerative data. The manuscript describes several benchmark studies using these methods in order to highlight the different analyses that can be performed. All methods are described in detail in the supplementary material. The software is freely available and can be very useful for the community.

Major points to consider:

1. cTI method: Trajectory-based PCA analysis is most-commonly used in single-cell gene expression approaches, however authors used it here for bulk gene expression patterns. How do they assure that the major principal components attribute to biological factors and not to technical issues (e.g. batches of data generation, lab influence etc.)? This is important because after grouping the method computes a disease score, and we want this disease score to reflect disease not technical issues. In addition, in the example of Fig. 2d: How much variance is explained by the different components?
2. Epidemic spreading model: From the model in Figure 3A-B) it is not really clear how the external inputs (drugs) could influence the changes in agents present. In the formula it says $\pm P_{\text{noise}}$ but how exactly is that defined?
3. Additionally, the system is composed of regions in the brain. Please give an estimate (e.g. in Methods) how many regions these are to get an idea of the size of the system.
4. How do the authors envision to incorporate further data, e.g. single-cell transcriptomics? This could be added to Discussion.
5. Another point for Discussion would be to describe the room for improvements. What is still missing and what are planned improvements?
6. I wasn't able to access the web page with the registry-free link given in the manuscript under "Note for Reviewers". I tried it with Firefox and Chrome browsers but in both cases I saw a "dead" background image with the message "Unspecified share exception". Thus, I didn't proceed to test the tool. Please check.

Minor points:

1. Page 7, Figure 2: Explain AD, HD and HC in the legend.
2. Page 10, Figure 3: Explain MP in the legend.
3. Page 10, 17: The model in Figure 3 is different from the model in the Methods (P_i instead of S_i , clearance rate ρ in the figure and δ in Methods). Please unify annotation. Check also the other models in the Figures and the Methods.
4. Page 22: References 12 and 33 are the same.

Reviewer #2 (Remarks to the Author):

The authors describe a new tool they've developed that is called NeuroPM-box, which integrates multimodal data, including molecular, histopathological, neuroimaging, and clinical neuroscience data. The tool allows users to perform analytical modeling, and they validate these methods on various data sets that include a large synthetic dataset, 911 in-vivo, and 736 post-mortem neurodegenerative data. They cite other studies that have successfully validated the tools and analytical methods included in their software; it would have been useful if the authors had expanded this part in the text to elaborate. They also cite some previous work from the community in terms of sharing integrative analytical modeling of these types of data, but this part of the introduction needs to be expanded so that the readers know what has been done, what is missing in this area, and how their tool fills in those gaps.

Their toolbox is open access and works across various platforms to help ensure it will be more widely used. They provide a schematic that explains the software workflow, but parts of this are a bit confusing and require more explanation.

The authors point out that this is a post-processing analytic software, but I wonder if they could link the tool with some preprocessing tools to make it easier for researchers to do both more easily in one place. Also the authors state that they are continuously developing new methods for this toolbox, but the details about future work is missing.

Overall, the work is very interesting and clearly a useful tool for many neuroscience researchers.

Reviewers' comments and replies:

Reviewer #1: In their manuscript Iturria-Medina et al. describe NeuroPM-box, a software that handles and unifies multiple heterogeneous data related to brain research. The different tools (e.g. trajectory-based gene expression analysis, spreading models) have been previously published. Additions and improvements of the published methods are highlighted. The methods were tested on several large data sets including simulated and experimental neurodegenerative data. The manuscript describes several benchmark studies using these methods in order to highlight the different analyses that can be performed. All methods are described in detail in the supplementary material. The software is freely available and can be very useful for the community. Points to consider:

Comment 1. cTI method: Trajectory-based PCA analysis is most-commonly used in single-cell gene expression approaches, however authors used it here for bulk gene expression patterns. How do they assure that the major principal components attribute to biological factors and not to technical issues (e.g. batches of data generation, lab influence etc.)? This is important because after grouping the method computes a disease score, and we want this disease score to reflect disease not technical issues. In addition, in the example of Fig. 2d: How much variance is explained by the different components?

Reply: Excellent comment, thank you for the opportunity to clarify these points. The reviewer is right, as mentioned in the original cTI article (Iturria-Medina et al., 2020, Brain 143, 1–13), this method was inspired by many other trajectory inference approaches developed and validated for single-cell analysis. Indeed, it is also critical to control by confounding factors. Thus, to further clarify the definition and application of the cTI method, in the revised manuscript version we have:

- 1) added an additional step on the algorithm's description (*online Methods, cTI definition* subsection): "Optional data adjustment for confounding factors. This step is strongly recommended for experimental data in which different conditions (e.g. technical procedures) may affect the quantitative comparison of observations and subsequent identification of relevant biological components. For instance, in this study before applying the cTI approach, each gene transcript's activity was adjusted for relevant covariates using robust additive linear models with pair-wise interactions (Street et al., 1988). Specifically, Dataset 1 GE (HBTRC) was adjusted for postmortem interval (PMI) in hours, age, gender and educational level. Dataset 2 GE (ADNI) was controlled for RIN, Plate Number, age, gender and educational level."
- 2) specified the variance explained by the different components. Specifically, adding (Fig. 2d): "This [cPCA dimensionality reduction] allows each subject to be represented in a reduced *n-dimensional* disease-associated space where the corresponding position reflects his/her pathological state (proximity to the bottom-left corner implies a pathology-free state; conversely, the top-right corner implies advanced pathology). For instance, when analyzing the GE data from the HBTRC's highly heterogeneous population (including HC, LOAD and HD subjects, total N=736), the high-dimensional data was reduced to seven contrasted PCs [cPCs] capturing up to 97.5% of the population variance (and individually explaining 38.73%, 19.91%, 16.18%, 8.46%, 5.85%, 5.50% and 2.87% of the variance, respectively). Notice that, for visualization simplicity, here were only represented the first three cPCs, but the quantitative analysis considers all the identified cPCs. Within this cPCs space, each

subject is automatically assigned to a disease-trajectory that represents a subpopulation of subjects potentially following a common disease variant (see *Methods*)...”

- 3) in line with this comment and # 4, we have added an associated discussion about the potential incorporation of other data modalities (e.g. single-cell RNA): “It is important to emphasize that, although the cTI results presented in this article are based on specific data types (bulk transcriptomics, histopathological data), there are not restrictions in the kind and number of data modalities that can be analyzed with this technique. Single-cell transcriptomic analysis is equally feasible with the current cTI implementation, potentially allowing the direct comparison with several trajectory inference methods originally proposed for such data type (Briggs et al., 2018; Cannoodt et al., 2016; Magwene et al., 2003; Welch et al., 2016). Furthermore, in complementary analyses (data not shown) we have recently confirmed the cTI’s capacity to concurrently integrate different data modalities (multi-omics molecular, multimodal neuroimaging, and/or several cognitive/behavioral/clinical evaluations), which is the main focus of our coming studies.”

Finally, with regards the adjustment of confounding factors, it is important to mention that one of the main attributes of the cTI method is its capacity to account by across-population effects that are common to both the background and target groups. The intrinsic dimensionality reduction technique (contrastive-PCA; Abid et al., 2018, *Nat. Commun.* 9, 1–24), focuses on detecting enriched patterns in the population of interest while adjusting by confounding components in the background population (e.g. remaining concurrent aging, gender and/or technical effects). We observed that this technique (cPCA) was significantly more sensitive to detecting disease progression than other popular dimensionality reduction methods (i.e. PCA and UMAP; Fig. S1 in Brain 143, 1–13, 2020), supporting the key advantage of considering the enriched patterns in the population of interest relative to the background dataset.

Comment 2. Epidemic spreading model: From the model in Figure 3A-B) it is not really clear how the external inputs (drugs) could influence the changes in agents present. In the formula it says +- P_{noise} but how exactly is that defined?

Reply: Thank you for this important observation. Although the ESM approach was initially defined by our group to consider external inputs and/or noise, the current software implementation doesn’t consider these components. Consequently, we have removed such effects/terms from the corresponding ESM text/descriptions and the equations in Figures 3A-B, while have added a note in the *Discussion* section about our goal to incorporate external inputs in future ESM-software versions (“Similar that for the MCM approach, inclusion of spatial molecular information and estimation of external inputs effects in the ESM are planned to be incorporated in future software versions”). We have made clear that, currently, consideration of external inputs is only implemented for the MCM approach (as specified in Figures 4A-B and mathematical model description in *Methods*).

Comment 3. Additionally, the system is composed of regions in the brain. Please give an estimate (e.g. in *Methods*) how many regions these are to get an idea of the size of the system.

Reply: As mentioned in *Methods (Example Dataset 2)*, for the first six mentioned brain imaging modalities, representative regional values were calculated for 78 regions covering all the grey

matter (Klein and Tourville, 2012). For further clarification, as suggested by the reviewer, we have also specified now the number of regions and corresponding parcellation when describing ESM and MCM approaches.

Comment 4. How do the authors envision to incorporate further data, e.g. single-cell transcriptomics? This could be added to Discussion.

Reply: Thank you for this comment. As mentioned in the reply to comment 1, in the revised manuscript we have added an associated discussion about the potential analysis of other data modalities (e.g. epigenetics, single-cell RNA): “It is important to emphasize that, although the cTI results presented in this article are based on specific data types (bulk transcriptomics, histopathological data), there are not restrictions in the kind and number of data modalities that can be analyzed with this technique. Single-cell transcriptomic analysis is equally feasible with the current cTI implementation, potentially allowing the direct comparison with several trajectory inference methods originally proposed for such data type (Briggs et al., 2018; Cannoodt et al., 2016; Magwene et al., 2003; Welch et al., 2016). Furthermore, in complementary analyses (data not shown) we have recently confirmed the cTI’s capacity to concurrently integrate different data modalities (molecular multi-omics, multimodal neuroimaging, and/or several cognitive/behavioral/clinical evaluations), which will be the main focus of our next studies in neurodegeneration.”

Comment 5. Another point for Discussion would be to describe the room for improvements. What is still missing and what are planned improvements?

Reply: Excellent comment, thanks. Motivated by it, we have added further details about coming improvements, including new methods that will be incorporated. For instance, in the Discussion section, now it can be read: “*NeuroPM-box* is a *long-term*, ongoing initiative. All of the tools included are under continuous development, particularly in terms of improving their numerical optimization (an open-ended field in research) and the interpretation/visualization of results. New tools and methods are also under development, with the goal of further integrating multiscale and multimodal neuroscience research. Future methodological additions will focus on continue bridging molecular, brain macroscopical factors (e.g. neuroimaging-derived biomarkers) and clinical data. For instance, we are in the process of incorporating a novel gene-expression informed MCM approach (Adewale et al., 2021; Adewale and Iturria-Medina, 2020), which proposes a general formulation that integrates whole-brain transcriptomic data of hundreds of landmark genes with multiple neuroimaging-derived biological factors (i.e. amyloid, metabolic and tau PET; vascular, functional, and structural MRI) and individual cognitive/clinical information. This unifying method, successfully validated on healthy aging and AD populations, concurrently accounts for the direct (causal) influence of hundreds of genes on regional macroscopic multifactorial effects, the pathological spreading of the ensuing aberrations (tau, amyloid) across axonal and vascular networks, and the resultant effects of these alterations on cognitive/clinical integrity. A similar multiscale brain model integrating neurotransmitter receptor densities with multimodal neuroimaging is also under development (Khan et al., 2020). Similar that for the MCM approach, inclusion of spatial molecular information and estimation of external inputs effects in the ESM are planned to be incorporated in future software versions.”

In addition, also in the *Discussion* section, it can be read: “Most of *NeuroPM-box*’s optimization algorithms have been implemented to minimize computational time. For instance, cTI can analyze thousands of subjects and large-scale omics data in just a few minutes (...) However, analyzing

hundreds of subjects with multimodal longitudinal imaging data from a regular workstation could take a few days (depending on the number of modalities, brain regions and time points; see Text S1). We are planning to upload the software to popular High-Performance Computing (HPC) portals, such as, The Neuroscience Gateway (NSG, <http://www.nsgportal.org>) and CBRAIN (<http://www.cbrain.ca>). Finally, to increase the software's generalizability, we are also working to extend data input to popular organizational formats, including the Brain Imaging Data Structure (BIDS) standard (Gorgolewski et al., 2016)."

Comment 6. I wasn't able to access the web page with the registry-free link given in the manuscript under "Note for Reviewers". I tried it with Firefox and Chrome browsers but in both cases I saw a "dead" background image with the message "Unspecified share exception". Thus, I didn't proceed to test the tool. Please check.

Reply: We sincerely apologize for this. While we have verified that the previous link (in our center's server) works well for multiple users, to avoid any potential issue, a new link for the reviewer is provided in the modified manuscript version. Please use this link (password is not required):

<https://box.bic.mni.mcgill.ca/s/WkPuEPVtQ2tpcO5>

If, for any reason, this link in our server can't be accessed by the reviewer, please access a mirrored version on a Google Drive folder:

https://drive.google.com/drive/folders/1PwQs0bREyAxIDXBRY4WBuur_W0rxkkNm?usp=sharing

Comment 7. Page7, Figure 2: Explain AD, HD and HC in the legend.

Reply: Thanks. Appropriate changes were made.

Comment 8. Page 10, Figure 3: Explain MP in the legend.

Reply: Thanks. Done.

Comment 9. Page 10, 17: The model in Figure 3 is different from the model in the Methods (P_i instead of S_i , clearance rate ρ in the figure and δ in Methods). Please unify annotation. Check also the other models in the Figures and the Methods.

Reply: Thank you for spotting this. We unified the annotation across the whole manuscript (replacing Figure 3 and revising the corresponding mathematical descriptions).

Comment 10. Page 22: References 12 and 33 are the same.

Reply: Thanks. Appropriate changes were made.

Reviewer #2: The authors describe a new tool they've developed that is called NeuroPM-box, which integrates multimodal data, including molecular, histopathological, neuroimaging, and clinical neuroscience data. The tool allows users to perform analytical modeling, and they validate these methods on various data sets that include a large synthetic dataset, 911 in-vivo, and 736 post-mortem neurodegenerative data. They cite other studies that have successfully validated the tools and analytical methods included in their software.

Comment 1: It would have been useful if the authors had expanded this part in the text to elaborate. They also cite some previous work from the community in terms of sharing integrative analytical modeling of these types of data, but this part of the introduction needs to be expanded so that the readers know what has been done, what is missing in this area, and how their tool fills in those gaps.

Reply: Thank you for this comment. In the revised manuscript version (*Introduction* and *Discussion* sections), we have added further clarification about critical missing aspects in this research area, motivating the development of a user-friendly multi-tool software for multiscale multimodal neuroscience data analysis.

For instance, in Introduction-page 2, it can be read: “The neuroinformatic field is similarly devoted to the development of analytical and computational models for the sharing, integration, and analysis of multimodal neuroscience data (Gaiteri et al., 2018; Mostafavi et al., 2018; Wu et al., 2020; Young et al., 2018). However, despite their potential to provide a better understanding of complex neuropathological processes and the individually-tailored selection of treatments, most associated methods (e.g. for separated or integrated molecular–neuroimaging analysis, data-driven patients stratification, or intra-brain spreading of pathological alterations) remain difficult to apply even when computational codes are shared, usually requiring advanced programming/technical expertise and, in many cases, even the collaboration of the developers. Simply put, vital user-friendly open-access tools for both multiscale and multifactorial brain research are still lacking. Their absence is accentuated by the accelerated development of innovative approaches requiring these types of tools. This contributes to statistical inconsistencies, consumes valuable research funding, and remains a major impediment to reproducibility in research.”

In Discussion-page15, is mentioned: “*NeuroPM-box* allows separated and combined analysis of data derived from molecular screening (transcriptomics, proteomics, epigenomics), histopathology (postmortem neuropathology), molecular imaging (amyloid, tau PET), macroscopic MRI, and cognitive/clinical evaluations. Most available packages focus exclusively on molecular (Bézioux et al., 2020; Bosco et al., 2019; Trapnell et al., 2014) or brain imaging (Fischl, 2012; Nichols et al., 2006; Sherif et al., 2014; Smith et al., 2004; Tournier et al., 2019) analysis, not on their combined analysis, which *NeuroPM-box* is specifically designed to address. Moreover, no other user-friendly software includes models for characterizing the intra-brain spreading of alteration effects (e.g., connectome-mediated tau and amyloid propagation as characterized by ESM and MCM) or for identifying individual therapeutic needs based on dynamical system analysis and control theory (e.g., pTIF). Although some validated computational codes for biomarkers-based patient stratification in the neurological context have been shared (Park et al., 2017; Young et al., 2018), the user requires programming or technical skills to apply them (...) it (*NeuroPM-box*) is deliberately designed to be a post-processing analytic software, not a preprocessing package, for which many excellent free software already exist (e.g., GEPAS, Bioconductor, SPM, FSL, ANTS, CIVET, FreeSurfer, MRtrix3, DSI studio, BrainSuite).

Consequently, basic molecular and imaging preprocessing (imaging registration, brain parcellation, quality control) should be completed beforehand.”

Comment 2: Their toolbox is open access and works across various platforms to help ensure it will be more widely used. They provide a schematic that explains the software workflow, but parts of this are a bit confusing and require more explanation.

Reply: Thank you for detecting this. Accordingly, we have added further description on the schematic Figure 1’s legend. In addition to the previous text, now it can be read: “E) Practical guidelines for methods users (available methods are further described in Table 1 and subsections below). Essentially, the analytical methods belong to two main categories, Empirical and Mechanistic. The former is purely data-driven and focus on identifying and interpreting intrinsic patterns in the data without making strong *a-priori* biological assumptions. Specifically, the included algorithm (see *contrasted Trajectory Inference* subsection and summary on Table 1) provide individualized quantitative scores reflective of disease progression and assign each subject to distinctive subpopulations (tentatively reflecting different disease sub-trajectories). Any type of quantitative data can be used as input (e.g. transcriptomic, proteomic, histopathological, metabolomics, multimodal imaging, clinical), while each data-feature’s contribution to the subjects’ final stratification is quantified, revealing the most informative features (e.g. specific genes, brain regions, clinical evaluations) and associated data modalities (e.g. RNA, imaging, clinical). However, the user should avoid performing causal interpretations based on empirical modeling, because the intrinsic limitation to distinguish between direct and indirect biological effects. Mechanistic models, by the contrary, aims to decode cause-effects in terms of biological factors alterations spreading through physical brain connections and/or synergistic factor-factor interactions contributing to spatiotemporal brain reorganization. The two implemented generative models focus on uni-modal or multi-modal imaging data, i.e. ESM considers the intra-brain spreading of a unique biological factor measured with an specific imaging modality (e.g. tau PET, or amyloid- β PET), while MCM considers the direct (causal) interactions and concurrent intra-brain spreading of multiple biological factors’ alterations quantified with different imaging modalities (tau, amyloid- β and glucose metabolism PET; cerebrovascular flow, functional activity indicators and structural atrophy measured with MRI). Notably, Mechanistic approaches (ESM, MCM, pTIF) can be informed by the Empirical data-driven outputs (cTI stratification), allowing the incorporation of a wide range of possible multi-scale biological information (e.g. molecular and clinical stages and subtypes) on the imaging-based generative brain models (see “cTI \rightarrow ESM, MCM, pTIF” method on Table 1). Finally, personalized causal brain models identified by the multi-modal Mechanistic approach (see MCM) can be interrogated to identify individual therapeutic needs in terms of biological deformations required to stop/revert factors-specific (imaging modalities) alterations or clinical deterioration (see pTIF on Table 1 and subsequent subsection).”

Comment 3: The authors point out that this is a post-processing analytic software, but I wonder if they could link the tool with some preprocessing tools to make it easier for researchers to do both more easily in one place. Also, the authors state that they are continuously developing new methods for this toolbox, but the details about future work is missing.

Reply: Thank you for this comment. Motivated by it, we have added further details about coming improvements, including new methods that will be incorporated. For instance, in the Discussion section, now it can be read: “*NeuroPM-box* is a *long-term*, ongoing initiative. All of the tools

included are under continuous development, particularly in terms of improving their numerical optimization (an open-ended field in research) and the interpretation/visualization of results. New tools and methods are also under development, with the goal of further integrating multiscale and multimodal neuroscience research. Future methodological additions will focus on continue bridging molecular, brain macroscopical factors (e.g. neuroimaging-derived biomarkers) and clinical data. For instance, we are in the process of incorporating a novel gene-expression informed MCM approach (Adewale et al., 2021; Adewale and Iturria-Medina, 2020), which proposes a general formulation that integrates whole-brain transcriptomic data of hundreds of landmark genes with multiple neuroimaging-derived biological factors (i.e. amyloid, metabolic and tau PET; vascular, functional, and structural MRI) and individual cognitive/clinical information. This unifying method, successfully validated on healthy aging and AD populations, concurrently accounts for the direct (causal) influence of hundreds of genes on regional macroscopic multifactorial effects, the pathological spreading of the ensuing aberrations (tau, amyloid) across axonal and vascular networks, and the resultant effects of these alterations on cognitive/clinical integrity. A similar multiscale brain model integrating neurotransmitter receptor densities with multimodal neuroimaging is also under development (Khan et al., 2020). Similar that for the MCM approach, inclusion of spatial molecular information and estimation of external inputs effects in the ESM are planned to be incorporated in future software versions.”

In addition, also in the *Discussion* section, it can be read: “Most of *NeuroPM-box*’s optimization algorithms have been implemented to minimize computational time. For instance, cTI can analyze thousands of subjects and large-scale omics data in just a few minutes (...) However, analyzing hundreds of subjects with multimodal longitudinal imaging data from a regular workstation could take a few days (depending on the number of modalities, brain regions and time points; see Text S1). We are planning to upload the software to popular High-Performance Computing (HPC) portals, such as, The Neuroscience Gateway (NSG, <http://www.nsgportal.org>) and CBRAIN (<http://www.cbrain.ca>). Finally, to increase the software’s generalizability, we are also working to extend data input to popular organizational formats, including the Brain Imaging Data Structure (BIDS) standard (Gorgolewski et al., 2016).”

Comment 4: Overall, the work is very interesting and clearly a useful tool for many neuroscience researchers.

Reply: Thank you very much, we sincerely appreciate the positive feedback.

References:

- Adewale, Q., Iturria-Medina, Y., 2020. Gene- neuroimaging brain model decodes neuropathological mechanisms in Alzheimer's disease. *Alzheimer's Dement.* 16, e047429. doi:<https://doi.org/10.1002/alz.047429>
- Adewale, Q., Khan, A.F., Carbonell, F., Iturria-Medina, Y., ADNI, 2021. Integrated Transcriptomic and Neuroimaging Brain Model Decodes Biological Mechanisms in Aging and Alzheimer's Disease. *eLife* (under-revision). medRxiv Prepr. <https://doi.org/10.1101/2021.02.23.21252283>.
- Bézieux, H.R. De, Street, K., Saelens, W., Saeys, Y., Dudoit, S., Clement, L., 2020. Trajectory-based differential expression analysis for single-cell sequencing data. *Nat. Commun.* 1–13. doi:10.1038/s41467-020-14766-3
- Bosco, G. Lo, Guan, J., Zhou, S., Gorban, A.N., Bauer, D.E., Aryee, M.J., Langenau, D.M., Zinovyev, A., Buenrostro, J.D., Yuan, G., Pinello, L., 2019. Single-cell trajectories reconstruction, exploration and mapping of omics data with STREAM. *Nat. Commun.* 10:1903. doi:10.1038/s41467-019-09670-4
- Briggs, J.A., Weinreb, C., Wagner, D.E., Megason, S., Peshkin, L., Kirschner, M.W., Klein, A.M., 2018. The dynamics of gene expression in vertebrate embryogenesis at single-cell resolution. *Science* (80-.). 360. doi:10.1126/science.aar5780
- Cannoodt, R., Saelens, W., Saeys, Y., 2016. Computational methods for trajectory inference from single-cell transcriptomics. *Eur. J. Immunol.* 46, 2496–2506. doi:10.1002/eji.201646347
- Fischl, B., 2012. Freesurfer. *Neuroimage* 62, 774–781.
- Gaiteri, C., Dawe, R., Mostafavi, S., Blizinsky, K.D., Tasaki, S., Komashko, V., Yu, L., Wang, Y., Schneider, J.A., Arfanakis, K., de Jager, P.L., Bennett, D.A., 2018. Gene expression and DNA methylation are extensively coordinated with MRI-based brain microstructural characteristics. *Brain Imaging Behav.* 1–10. doi:10.1007/s11682-018-9910-4
- Gorgolewski, K.J., Auer, T., Calhoun, V.D., Craddock, R.C., Das, S., 2016. The brain imaging data structure , a format for organizing and describing outputs of neuroimaging experiments 1–9.
- Khan, A.F., Palomero-Gallagher, N., Zilles, K., Iturria-Medina, Y., 2020. Whole brain generative model identifies neurotransmitter alterations underlying Alzheimer's disease progression. *Alzheimer's Dement.* 16, e041193. doi:<https://doi.org/10.1002/alz.041193>
- Klein, A., Tourville, J., 2012. 101 Labeled Brain Images and a Consistent Human Cortical Labeling Protocol. *Front. Neurosci.* 6, 171. doi:10.3389/fnins.2012.00171
- Magwene, P.M., Kim, P.L., Junhyong, 2003. Reconstructing the temporal ordering of biological samples using microarray data. *Bioinformatics Vol.* 19, 842–850.
- Mostafavi, S., Gaiteri, C., Sullivan, S.E., White, C.C., Tasaki, S., Xu, J., Taga, M., Klein, H., Patrick, E., Komashko, V., McCabe, C., Smith, R., Bradshaw, E.M., Root, D.E., Regev, A., Yu, L., Chibnik, L.B., Schneider, J.A., Young-pearse, T.L., Bennett, D.A., Jager, P.L. De, 2018. A molecular network of the aging human brain provides insights into the pathology and cognitive decline of Alzheimer's disease. *Nat. Neurosci.* 21. doi:10.1038/s41593-018-0154-9

- Nichols, T., Penny, W., Friston, K., Ashburner, J., Kiebel, S., 2006. *Statistical Parametric Mapping: The Analysis of Functional Brain Images*, 1st ed. Academic Press.
- Park, J.-Y., Na, H.K., Kim, et al., 2017. Robust Identification of Alzheimer's Disease subtypes based on cortical atrophy patterns. *Sci. Rep.* 7, 43270. doi:10.1038/srep43270
- Sherif, T., Rioux, P., Rousseau, M.-E., Kassis, N., Beck, N., Adalat, R., Das, S., Glatard, T., Evans, A.C., 2014. CBRAIN: a web-based, distributed computing platform for collaborative neuroimaging research. *Front. Neuroinform.* 8, 1–13. doi:10.3389/fninf.2014.00054
- Smith, S.M., Jenkinson, M., Woolrich, M.W., Beckmann, C.F., Behrens, T.E.J., Johansen-Berg, H., Bannister, P.R., Luca, M. De, Drobnjak, I., Flitney, D.E., Niazy, R., Saunders, J., Vickers, J., Zhang, Y., Stefano, N. De, Brady, J.M., Matthews, P.M., 2004. Advances in functional and structural MR image analysis and implementation as FSL. *Neuroimage* 23(1), S208–S219.
- Street, J.O., Carroll, R.J., Ruppert, D., 1988. A Note on Computing Robust Regression Estimates via Iteratively Reweighted Least Squares. *Am. Stat.* 42, 152–154.
- Tournier, J.-D., Smith, R., Raffelt, D., Tabbara, R., Dhollander, T., Pietsch, M., Christiaens, D., Jeurissen, B., Yeh, C.-H., Connelly, A., 2019. MRtrix3: A fast, flexible and open software framework for medical image processing and visualisation. *Neuroimage* 202, 116137.
- Trapnell, C., Cacchiarelli, D., Grimsby, J., Pokharel, P., Li, S., Morse, M., Lennon, N.J., Livak, K.J., Mikkelsen, T.S., Rinn, J.L., 2014. The dynamics and regulators of cell fate decisions are revealed by pseudotemporal ordering of single cells. *Nat. Biotechnol.* 32, 381–386. doi:10.1038/nbt.2859
- Welch, J.D., Hartemink, A.J., Prins, J.F., 2016. SLICER: Inferring branched, nonlinear cellular trajectories from single cell RNA-seq data. *Genome Biol.* 17, 1–15. doi:10.1186/s13059-016-0975-3
- Wu, W., Zhang, Y., Jiang, J., Lucas, M. V, Fonzo, G.A., Rolle, C.E., Cooper, C., Chin-fatt, C., Krepel, N., Cornelissen, C.A., Wright, R., Toll, R.T., Trivedi, H.M., Monuszko, K., Caudle, T.L., Sarhadi, K., Jha, M.K., Trombello, J.M., Deckersbach, T., Adams, P., Mcgrath, P.J., Weissman, M.M., Fava, M., Pizzagalli, D.A., Arns, M., Trivedi, M.H., Etkin, A., 2020. An electroencephalographic signature predicts antidepressant response in major depression. *Nat. Biotechnol.* 38. doi:10.1038/s41587-019-0397-3
- Young, A.L., Marinescu, R.-V. V, Oxtoby, N.P. et al, 2018. Uncovering the heterogeneity and temporal complexity of neurodegenerative diseases with Subtype and Stage Inference. *Nat. Commun.* 9:4273. doi:10.1101/236604

REVIEWERS' COMMENTS:

Reviewer #1 (Remarks to the Author):

The authors have addressed my comments sufficiently and adequately. I have no further comments and support publication of the manuscript.

Reviewer #2 (Remarks to the Author):

The comments were addressed very well, and the revisions have greatly improved the manuscript.